# GENERATING PHYSICAL DYNAMICS UNDER PRIORS

**Zihan Zhou**[1], **Xiaoxue Wang**[2], **Tianshu Yu**[1*]
[1]School of Data Science, The Chinese University of Hong Kong
[2]ChemLex Technology Co., Ltd.
`zihanzhou1@link.cuhk.edu.cn, wxx@chemlex.tech`
`yutianshu@cuhk.edu.cn`

## ABSTRACT

Generating physically feasible dynamics in a data-driven context is challenging, especially when adhering to *physical priors* expressed in specific equations or formulas. Existing methodologies often overlook the integration of "physical priors", resulting in violation of basic physical laws and suboptimal performance. In this paper, we introduce a novel framework that seamlessly incorporates physical priors into diffusion-based generative models to address this limitation. Our approach leverages two categories of priors: 1) distributional priors, such as roto-translational invariance, and 2) physical feasibility priors, including energy and momentum conservation laws and PDE constraints. By embedding these priors into the generative process, our method can efficiently generate physically realistic dynamics, encompassing trajectories and flows. Empirical evaluations demonstrate that our method produces high-quality dynamics across a diverse array of physical phenomena with remarkable robustness, underscoring its potential to advance data-driven studies in AI4Physics. Our contributions signify a substantial advancement in the field of generative modeling, offering a robust solution to generate accurate and physically consistent dynamics.

## 1 INTRODUCTION

The generation of physically feasible dynamics is a fundamental challenge in the realm of data-driven modeling and AI4Physics. These dynamics, driven by Partial Differential Equations (PDEs), are ubiquitous in various scientific and engineering domains, including fluid dynamics (Kutz, 2017), climate modeling (Rasp et al., 2018), and materials science (Choudhary et al., 2022). Accurately generating such dynamics is crucial for advancing our understanding and predictive capabilities in these fields (Bzdok & Ioannidis, 2019). Recently, generative models have revolutionized the study of physics by providing powerful tools to simulate and predict complex systems.

**Generative v.s. discriminative models.** Even when high-performing discriminative models for dynamics are available such as finite elements (Zhang et al., 2021; Uriarte et al., 2022), finite difference (Lu et al., 2021; Salman et al., 2022), finite volume (Ranade et al., 2021) or physics-informed neural networks (PINNs) (Raissi et al., 2019), generative models are crucial in machine learning for their ability to capture the full data distribution, enabling more effective data synthesis (de Oliveira et al., 2017), anomaly detection (Finke et al., 2021), and semi-supervised learning (Ma et al., 2019). They enhance robustness and interpretability by modeling the joint distribution of data and labels, offering insights into unseen scenarios (Takeishi & Kalousis, 2021). Generative models are also pivotal in creative domains, such as drug discovery (Lavecchia, 2019), where they enable the creation of novel data samples.

**Challenge.** However, the intrinsic complexity and high-dimensional nature of physical dynamics pose significant challenges for traditional learning systems. Recent advancements in generative modeling, particularly diffusion-based generative models (Song et al., 2020), have shown promise in capturing complex data distributions. These models iteratively refine noisy samples to match the target distribution, making them well-suited for high-dimensional data generation. Despite their success, existing approaches often overlook the incorporation of "physical priors" expressed in spe-

---

*corresponding author

cific equations or formulas, which are essential for ensuring that the generated dynamics adhere to fundamental physical laws.

**Solution.** In this work, we propose a novel framework that integrates priors into diffusion-based generative models to generate physically feasible dynamics. Our approach leverages two types of priors: *Distributional priors*, including roto-translational invariance and equivariance, ensure that models capture the intrinsic properties of the data rather than their specific representations; *Physical feasibility priors*, including energy and momentum conservation laws and PDE constraints, enforce the adherence to fundamental physical principles, thus improving the quality of generated dynamics.

The integration of priors into the generative process is a complex task that necessitates a deep understanding of the relevant mathematical and physical principles. Unlike predictive tasks, where the objective is to estimate a *specific ground-truth value* $x_0$, diffusion generative models aim to characterize a *full ground-truth distribution* $\nabla_x \log q_t(x_t)$ or $\mathbb{E}[x_0 \mid x_t]$ (notation in Equation 1). This fundamental difference complicates the direct application of priors based on ground-truth values to the output of generative models. In this work, we propose a framework to address this challenge by effectively embedding priors within the generative model's output distribution. By incorporating these priors into a diffusion-based generation framework, our approach can efficiently produce physically plausible dynamics. This capability is particularly useful for studying physical phenomena where the governing equations are too complex to be learned purely from data.

Figure 1: Animated visualization of generated samples of shallow water dynamics, showcasing the variations over time. Use the latest version of Adobe Acrobat Reader to view.

**Results.** Empirical evaluations of our method demonstrate its effectiveness in producing high-quality dynamics across a range of physical phenomena. Our approach exhibits high robustness and generalizability, making it a promising tool for the data-driven study of AI4Physics. In Fig. 1, we provide a generated sample of the shallow water dataset (Martínez-Aranda et al., 2018). The generated dynamics not only capture the intricate details of the physical processes but also adhere to the fundamental physical laws, offering an accurate and reliable representation of underlying systems.

**Contribution.** In conclusion, our work presents a significant advancement in the field of data-driven generative modeling by introducing a novel framework that integrates physical priors into diffusion-based generative models. In all, our method **1**) improves the feasibility of generated dynamics, making them more aligned with physical principles compared to baseline methods; **2**) poses the solution to the longstanding challenge of generating physically feasible dynamics; **3**) paves the way for more accurate and reliable data-driven studies in various scientific and engineering domains, highlighting the potential of AI4Physics in advancing our understanding of complex physical systems.

## 2 PRELIMINARIES

In Appendix A, we present a comprehensive review of *Related Work*, specifically focusing on three key areas: generative methods for physics, score-based diffusion models, and physics-informed neural networks. This section aims to provide foundational knowledge for readers who may not be familiar with these topics. We recommend that those seeking to deepen their understanding of these areas consult this appendix.

### 2.1 DIFFUSION MODELS

Diffusion models generate samples following an underlying distribution. Consider a random variable $x_0 \in \mathbb{R}^n$ drawn from an unknown distribution $q_0$. Denoising diffusion probabilistic models (Song & Ermon, 2019; Song et al., 2020; Ho et al., 2020) describe a forward process $x_t, t \in [0, T]$ governed by an Ito stochastic differential equation (SDE)

$$\mathrm{d}x_t = f(t)x_t\mathrm{d}t + g(t)\mathrm{d}\mathbf{w}_t, \quad x_0 \sim q_0, \quad f(t) = \frac{\mathrm{d}\log\alpha_t}{\mathrm{d}t}, g^2(t) = \frac{\mathrm{d}\sigma_t^2}{\mathrm{d}t} - 2\frac{\mathrm{d}\log\alpha_t}{\mathrm{d}t}\sigma_t^2, \quad (1)$$

where $\mathbf{w}_t \in \mathbb{R}^n$ denotes the standard Brownian motion, and $\alpha_t$ and $\sigma_t$ are predetermined functions of $t$. This forward process has a closed-form solution of $q_t(\boldsymbol{x}_t \mid \boldsymbol{x}_0) = \mathcal{N}(\boldsymbol{x}_t \mid \alpha_t \boldsymbol{x}_0, \sigma_t^2 \mathbf{I})$ and has a corresponding reverse process of the probability flow ordinary differential equation (ODE), running from time $T$ to 0, defined as (Song et al., 2020)

$$\frac{\mathrm{d}\boldsymbol{x}_t}{\mathrm{d}t} = f(t)\boldsymbol{x}_t - \frac{1}{2}g^2(t)\nabla_{\boldsymbol{x}} \log q_t(\boldsymbol{x}_t), \quad \boldsymbol{x}_T \sim q_T(\boldsymbol{x}_T \mid \boldsymbol{x}_0) \approx q_T(\boldsymbol{x}_T). \tag{2}$$

The marginal probability densities $\{q_t(\boldsymbol{x}_t)\}_{t=0}^T$ of the forward SDE align with the reverse ODE (Song et al., 2020). This indicates that if we can sample from $q_T(\boldsymbol{x}_T)$ and solve Equation 2, then the resulting $\boldsymbol{x}_0$ will follow the distribution $q_0$. By choosing $\alpha_t \to 0$ and $\sigma_t \to 1$, the distribution $q_T(\boldsymbol{x}_T)$ can be approximated as a normal distribution. The score $\nabla_{\boldsymbol{x}} \log q_t(\boldsymbol{x}_t)$ can be approximated by a deep learning model. The quality of the generated samples is contingent upon the models' ability to accurately approximate the score functions (Kwon et al., 2022; Gao & Zhu, 2024). A more precise approximation results in a distribution that closely aligns with the distribution of the training set. To enhance model fit, incorporating priors of the distributions and physical feasibility into the models is advisable. Section 3 will elaborate on our methods for integrating distributional priors and physical feasibility priors, as well as the objectives for score matching.

## 2.2 INVARIANT DISTRIBUTIONS

An *invariant distribution* refers to a probability distribution that remains unchanged under the action of a specified group of transformations. These transformations can include operations such as translations, rotations, or other symmetries, depending on the problem domain. Formally, let $\mathcal{G}$ be a group of transformations. A distribution $q$ is said to be $\mathcal{G}$-*invariant* under the group $\mathcal{G}$ if for all transformations $\mathbf{G} \in \mathcal{G}$, we have $q(\mathbf{G}(\boldsymbol{x})) = q(\boldsymbol{x})$. Invariance under group transformations is particularly significant in modeling distributions that exhibit symmetries. For instance, in the case of 3D coordinates, invariance under rigid transformations—such as translations and rotations (SE(3) group)—is essential for spatial understanding (Zhou et al., 2024). Equivariant models are usually required to embed invariance. A function (or model) $\mathbf{f} : \mathbb{R}^n \to \mathbb{R}^n$ is said to be $(\mathcal{G}, \mathcal{L})$-*equivariant* where $\mathcal{G}$ is the group actions and $\mathcal{L}$ is a function operator, if for any $\mathbf{G} \in \mathcal{G}$, $\mathbf{f}(\mathbf{G}(\boldsymbol{x})) = \mathcal{L}(\mathbf{G})(\mathbf{f}(\boldsymbol{x}))$.

## 3 METHOD

In this study, we aim to investigate methodologies for enhancing the capability of diffusion models to approximate the targeted score functions. We have two primary objectives: **1**) To incorporate distributional priors, such as translational and rotational invariance, which will aid in selecting the appropriate model for training objective functions; **2**) To impose physical feasibility priors on the diffusion model, necessitating injection of priors to model's output of a distribution related to the ground-truth samples (specifically, $\nabla_{\boldsymbol{x}} \log q_t(\boldsymbol{x}_t)$ or $\mathbb{E}[\boldsymbol{x}_0 \mid \boldsymbol{x}_t]$). In this section, we consider the forward diffusion process given by Equation 1, where $\boldsymbol{x}_t = \alpha_t \boldsymbol{x}_0 + \sigma_t \boldsymbol{\epsilon}$, with $\boldsymbol{\epsilon} \sim \mathcal{N}(\mathbf{0}, \mathbf{I})$.

### 3.1 INCORPORATING DISTRIBUTIONAL PRIORS

In this section, we study the score function $\nabla_{\boldsymbol{x}} \log q_t(\boldsymbol{x}_t)$ for $\mathcal{G}$-invariant distributions. Understanding its corresponding properties can guide the selection of models with the desired equivariance, facilitating sampling from the $\mathcal{G}$-invariant distribution. In the following, we will assume that the sufficient conditions of Theorem 1 hold so that the marginal distributions $q_t$ are $\mathcal{G}$-invariant. The definitions of the terminologies and proof of the theorem can be found in Appendix F.1.

**Theorem 1** (Sufficient conditions for the invariance of $q_0$ to imply the invariance of $q_t$)**.** *Let $q_0$ be a $\mathcal{G}$-invariant distribution. If for all $\mathbf{G} \in \mathcal{G}$, $\mathbf{G}$ is volume-preserving diffeomorphism and isometry, and for all $0 < a < 1$, there exists $\mathbf{H} \in \mathcal{G}$ such that $\mathbf{H}(a\boldsymbol{x}) = a\mathbf{G}(\boldsymbol{x})$, then $q_t$ is also $\mathcal{G}$-invariant.*

**Property of score functions.** Let $q_t$ be a $\mathcal{G}$-invariant distribution. By the chain rule, we have $\nabla_{\boldsymbol{x}} \log q_t(\boldsymbol{x}_t) = \nabla_{\boldsymbol{x}} \log q_t(\mathbf{G}(\boldsymbol{x}_t)) = \frac{\partial \mathbf{G}(\boldsymbol{x}_t)}{\partial \boldsymbol{x}} \nabla_{\mathbf{G}(\boldsymbol{x})} \log q_t(\mathbf{G}(\boldsymbol{x}_t))$, for all $\mathbf{G} \in \mathcal{G}$. Hence,

$$\nabla_{\mathbf{G}(\boldsymbol{x})} \log q_t(\mathbf{G}(\boldsymbol{x}_t)) = \left(\frac{\partial \mathbf{G}(\boldsymbol{x}_t)}{\partial \boldsymbol{x}}\right)^{-1} \nabla_{\boldsymbol{x}} \log q_t(\boldsymbol{x}_t). \tag{3}$$

This implies that the score function of $\mathcal{G}$-invariant distribution is $(\mathcal{G}, \nabla^{-1})$-equivariant. We should use a $(\mathcal{G}, \nabla^{-1})$-equivariant model to predict the score function. The loss objective is given by

$$\mathcal{J}_{\text{score}}(\boldsymbol{\theta}) = \mathbb{E}_{t,\boldsymbol{x}_0,\boldsymbol{\epsilon}} \left[ w(t) \left\| \mathbf{s}_{\boldsymbol{\theta}}\left(\boldsymbol{x}_t, t\right) - \nabla_{\boldsymbol{x}} \log q_t(\boldsymbol{x}_t) \right\|^2 \right], \tag{4}$$

where $w(t)$ is a positive weight function and $\mathbf{s}_{\boldsymbol{\theta}}$ is a $(\mathcal{G}, \nabla^{-1})$-equivariant model. We will discuss the handling of the intractable score function $\nabla_{\boldsymbol{x}} \log q_t(\boldsymbol{x}_t)$ subsequently in Equation 6.

In the context of simulating physical dynamics, two distributional priors are commonly considered: SE(n)-invariance and permutation-invariance. They ensure that the learned representations are consistent with the fundamental symmetries of physical laws, including rigid body transformations and indistinguishability of particles, thereby enhancing the model's ability to generalize across different physical scenarios. The derivations for the following examples can be found in Appendix F.2.

**Example 1.** *(SE(n)-invariant distribution) If $q_0$ is an SE(n)-invariant distribution, then $q_t$ is also SE(n)-invariant. The score function of an SE(n)-invariant distribution is SO(n)-equivariant and translational-invariant.*

**Example 2.** *(Permutation-invariant distribution) If $q_0$ is a permutation-invariant, then $q_t$ is also permutation-invariant. The score function of a permutation-invariant distribution is permutation-equivariant.*

In the following, we will show that using such a $(\mathcal{G}, \nabla^{-1})$-equivariant model, we are essentially training a model that focuses on the intrinsic structure of data instead of their representation form.

**Equivalence class manifold for invariant distributions.** An *equivalence class manifold* (ECM) refers to the minimum subset of samples where all the rest elements are considered equivalent to one of the samples in this manifold (informal). For example, in three-dimensional space, coordinates that have undergone rotation and translation maintain their pairwise distances, which allows the use of a set of coordinates to represent all other coordinates with the same distance matrices, thereby forming an equivalence class manifold (see Appendix B for the formal definition and examples). By incorporating the invariance prior to the training set, we can construct ECM from the training set or a mini-batch of samples. The utilization of ECM enables the models to concentrate on the intrinsic structure of the data, thereby enhancing generalization and robustness to irrelevant variations. We assume that the distribution of $\boldsymbol{x}$ follows an $\mathcal{G}$-invariant distribution $q_t$. Let $\varphi$ map $\boldsymbol{x}_t$ to the corresponding point having the same intrinsic structure in ECM. Then there exists $\mathbf{G} \in \mathcal{G}$ such that $\mathbf{G}(\varphi(\boldsymbol{x}_t)) = \boldsymbol{x}_t$. Since $q_t$ is $\mathcal{G}$-invariant, we have $q_t(\boldsymbol{x}_t) = q_{\text{ECM}}(\varphi(\boldsymbol{x}_t)) \cdot p_{\text{Uniform}(\mathcal{G})}(\mathbf{G})$, where $q_{\text{ECM}}$ is defined on the domain of ECM. Taking the logarithm and derivative, we have $\nabla_{\varphi(\boldsymbol{x})} \log q_t(\boldsymbol{x}_t) = \nabla_{\varphi(\boldsymbol{x})} \log q_{\text{ECM}}(\varphi(\boldsymbol{x}_t))$. Note that $\nabla_{\boldsymbol{x}} \log q_t(\boldsymbol{x}_t) = \frac{\partial \varphi(\boldsymbol{x}_t)}{\partial \boldsymbol{x}} \nabla_{\varphi(\boldsymbol{x})} \log q_t(\boldsymbol{x}_t)$. Hence,

$$\nabla_{\boldsymbol{x}} \log q_t(\boldsymbol{x}_t) = \frac{\partial \varphi(\boldsymbol{x}_t)}{\partial \boldsymbol{x}} \nabla_{\varphi(\boldsymbol{x})} \log q_{\text{ECM}}(\varphi(\boldsymbol{x}_t)). \tag{5}$$

This implies that the score function of the $\mathcal{G}$-invariant distribution is closely related to the score function of the distribution in ECM. Such a result indicates that if we have a $(\mathcal{G}, \nabla^{-1})$-equivariant model that can predict the score functions in ECM, then, this model predicts the score functions for all other points closed under the group operation. We summarize this result in the following theorem whose proofs can be found in Appendix F.3.

**Theorem 2** (Equivalence class manifold representation). *If we have a $(\mathcal{G}, \nabla^{-1})$-equivariant model such that $\mathbf{s}_{\boldsymbol{\theta}}(\boldsymbol{x}_t, t) = \nabla_{\boldsymbol{x}} \log q_{\text{ECM}}(\boldsymbol{x}_t)$ almost surely on $\boldsymbol{x}_t \in \text{ECM}$, then we have $\mathbf{s}_{\boldsymbol{\theta}}(\boldsymbol{x}_t, t) = \nabla_{\boldsymbol{x}} \log q_t(\boldsymbol{x}_t)$ almost surely.*

**Objective for fitting the score function.** The score function $\nabla_{\boldsymbol{x}} \log q_t(\boldsymbol{x}_t)$ is generally intractable and we consider the objective for *noise matching* and *data matching* (Vincent, 2011; Song et al., 2020; Zheng et al., 2023), where objectives and optimal values are given by

$$\mathcal{J}_{\text{noise}}(\boldsymbol{\theta}) = \mathbb{E}_{t,\boldsymbol{x}_0,\boldsymbol{\epsilon}} \left[ w(t) \left\| \boldsymbol{\epsilon}_{\boldsymbol{\theta}}\left(\boldsymbol{x}_t, t\right) - \boldsymbol{\epsilon} \right\|^2 \right], \qquad \boldsymbol{\epsilon}_{\boldsymbol{\theta}}^*\left(\boldsymbol{x}_t, t\right) = -\sigma_t \nabla_{\boldsymbol{x}} \log q_t\left(\boldsymbol{x}_t\right); \tag{6a}$$

$$\mathcal{J}_{\text{data}}(\boldsymbol{\theta}) = \mathbb{E}_{t,\boldsymbol{x}_0,\boldsymbol{\epsilon}} \left[ w(t) \left\| \boldsymbol{x}_{\boldsymbol{\theta}}\left(\boldsymbol{x}_t, t\right) - \boldsymbol{x}_0 \right\|^2 \right], \qquad \boldsymbol{x}_{\boldsymbol{\theta}}^*\left(\boldsymbol{x}_t, t\right) = \frac{1}{\alpha_t} \boldsymbol{x}_t + \frac{\sigma_t^2}{\alpha_t} \nabla_{\boldsymbol{x}} \log q_t\left(\boldsymbol{x}_t\right). \tag{6b}$$

The diffusion objectives for both the noise predictor $\boldsymbol{\epsilon_\theta}$ and the data predictor $\boldsymbol{x_\theta}$ are intrinsically linked to the score function, thereby inheriting its characteristics and properties. However, the data predictor incorporates a term, $\frac{1}{\alpha_t}\boldsymbol{x}_t$, whose numerical range exhibits instability. This instability complicates the predictor's ability to inherit the straightforward properties of the score function. Therefore, to incorporate $\mathcal{G}$-invariance, it is advisable to employ noise matching, which is given by Equation 6a and $\boldsymbol{\epsilon_\theta}$ is $(\mathcal{G}, \nabla^{-1})$-equivariant, which is the property of the score function.

A specific instance of a distributional prior is defined by samples that conform to the constraints imposed by PDEs. In this context, the dynamics at any given spatial location depend solely on the characteristics of the system within its local vicinity, rather than on absolute spatial coordinates. Under these conditions, it is appropriate to employ translation-invariant models for both noise matching and data matching. Nevertheless, the samples in question exhibit significant smoothness. As a result, utilizing the noise matching objective necessitates that the model's output be accurate at every individual pixel. In contrast, applying the data matching objective only requires the model to produce smooth output values. Therefore, it is recommended to adopt the data matching objective for this purpose. The selection between data matching and noise matching plays a critical role in determining the quality of the generated samples. For detailed experimental results, refer to Sec. 4.3.

**Remark 1.** *In this section, we primarily explore the principle for incorporating distributional priors by selecting models with particular characteristics. Specifically:*

1. *When the distribution exhibits $\mathcal{G}$-invariance, a $(\mathcal{G}, \nabla^{-1})$-equivariant model should be employed alongside the **noise matching** objective (Equation 6a).*

2. *For samples that are subject to PDE constraints and exhibit high smoothness, the **data matching** objective (Equation 6b) is recommended.*

### 3.2 INCORPORATING PHYSICAL FEASIBILITY PRIORS

In this section, we explore how to incorporate physical feasibility priors such as physics laws and explicit PDE constraints into noise and data matching objectives in diffusion models. By Tweedie's formula (Efron, 2011; Kim & Ye, 2021; Chung et al., 2022), we have $\mathbb{E}[\boldsymbol{x}_0 \mid \boldsymbol{x}_t] = \frac{1}{\alpha_t}\left(\boldsymbol{x}_t + \sigma_t^2 \nabla_{\boldsymbol{x}} \log q_t(\boldsymbol{x}_t)\right)$. Hence,

$$\mathbb{E}[\boldsymbol{x}_0 \mid \boldsymbol{x}_t] = \frac{1}{\alpha_t}\left(\boldsymbol{x}_t - \sigma_t \boldsymbol{\epsilon}_{\boldsymbol{\theta}}^*(\boldsymbol{x}_t, t)\right), \quad \mathbb{E}[\boldsymbol{x}_0 \mid \boldsymbol{x}_t] = \boldsymbol{x}_{\boldsymbol{\theta}}^*(\boldsymbol{x}_t, t). \tag{7}$$

For both noise and data matching objectives, we are essentially training a model to approximate $\mathbb{E}[\boldsymbol{x}_0 \mid \boldsymbol{x}_t]$. A purely data-driven approach is often insufficient to capture the underlying physical constraints accurately. Therefore, similar to PINNs (Leiteritz & Pflüger, 2021), we incorporate an additional penalty loss $\mathcal{J}_{\mathcal{R}}$ into the objective function to enforce physical feasibility priors $\mathcal{R}(\boldsymbol{x}_0) = \boldsymbol{0}$ and set the loss objective to be $\mathcal{J}(\boldsymbol{\theta}) = \mathcal{J}_{\text{score}}(\boldsymbol{\theta}) + \lambda \mathcal{J}_{\mathcal{R}}(\boldsymbol{\theta})$, where $\mathcal{J}_{\text{score}}$ is the data matching or noise matching objectives and $\lambda$ is a hyperparameter to balance the diffusion loss and physical feasibility loss. We consider the data matching objective where $\boldsymbol{x_\theta}(\boldsymbol{x}_t, t)$ approximates $\mathbb{E}[\boldsymbol{x}_0 \mid \boldsymbol{x}_t]$. For noise matching models, we can transform the model's output by Equation 7. For general cases, we cannot directly add the constraints $\mathcal{R}(\boldsymbol{x}_0) = \boldsymbol{0}$ to the output of the diffusion model $\mathbb{E}[\boldsymbol{x}_0 \mid \boldsymbol{x}_t]$ due to the presence of Jensen's gap (Bastek et al., 2024), i.e., $\mathcal{R}(\mathbb{E}[\boldsymbol{x}_0 \mid \boldsymbol{x}_t]) \neq \mathbb{E}[\mathcal{R}(\boldsymbol{x}_0) \mid \boldsymbol{x}_t] = \boldsymbol{0}$. However, in some special cases, we can avoid dealing with this gap.

**Linear cases.** When the constraints are linear/affine functions, Jensen's gap equals 0. Hence, we can directly add the constraints to $\boldsymbol{x_\theta}(\boldsymbol{x}_t, t)$. We have $\mathcal{J}_{\mathcal{R}}(\boldsymbol{\theta}) = \mathbb{E}_{t, \boldsymbol{x}_0, \boldsymbol{\epsilon}}\left[w(t)\|\mathcal{R}(\boldsymbol{x_\theta}(\boldsymbol{x}_t, t))\|^2\right]$.

**Multilinear cases.** A function is called multilinear if it is linear in several arguments when the other arguments are fixed. Denote $\boldsymbol{x}_0 = \begin{bmatrix} \mathbf{u}_0 \\ \mathbf{v}_0 \end{bmatrix} \in \mathbb{R}^{m+n}, \mathbf{u}_0 \in \mathbb{R}^m, \mathbf{v}_0 \in \mathbb{R}^n$. When the constraints function is multilinear w.r.t. $\mathbf{u}_0$, we can write the constraints in the form of $\mathcal{R}(\boldsymbol{x}_0) = \mathbf{W}_0 \mathbf{u}_0 + \mathbf{b}_0 = \boldsymbol{0}$, where $\mathbf{W}_0$ and $\mathbf{b}_0$ are functions of $\mathbf{v}_0$. In this case, we can use the penalty loss as $\mathcal{J}_{\mathcal{R}}(\boldsymbol{\theta}) = \mathbb{E}_{t, \boldsymbol{x}_0, \boldsymbol{\epsilon}}[w(t)\|\mathbf{W}_0 \mathbf{u_\theta}(\boldsymbol{x}_t, t) + \mathbf{b}_0\|^2]$. Such a design is supported by the following theorem whose proof can be found in Appendx F.4.

**Theorem 3** (Multilinear Jensen's gap). *The optimizer for $\mathbb{E}_{t,\boldsymbol{x}_0,\boldsymbol{\epsilon}}[w(t)\|\mathbf{u}_{\boldsymbol{\theta}_1}(\boldsymbol{x}_t,t)-\mathbf{u}_0\|^2]$ is the reweighted optimizer of $\mathbb{E}_{t,\boldsymbol{x}_0,\boldsymbol{\epsilon}}[w(t)\|\mathbf{W}_0\mathbf{u}_{\boldsymbol{\theta}_2}(\boldsymbol{x}_t,t)+\mathbf{b}_0\|^2]$ with reweighted variable $\mathbf{W}_0^\top\mathbf{W}_0$.*

**Convex cases.** If the constraints $\mathcal{R}$ is convex, by Jensen's inequality, $\mathcal{R}\left(\mathbb{E}[\boldsymbol{x}_0\mid\boldsymbol{x}_t]\right)\leq\mathbb{E}[\mathcal{R}\left(\boldsymbol{x}_0\right)\mid\boldsymbol{x}_t]=\mathbf{0}$. Hence, $0=\|\mathbb{E}[\mathcal{R}\left(\boldsymbol{x}_0\right)\mid\boldsymbol{x}_t]\|^2\leq\|\mathcal{R}\left(\mathbb{E}[\boldsymbol{x}_0\mid\boldsymbol{x}_t]\right)\|^2$. When a data matching model is approximately optimized, directly applying constraints to the model's output minimizes the upper bound of the constraints on $\boldsymbol{x}_0$. The upper bound of the Jensen's gap is related the absolute centered moment $\sigma_p=\sqrt[p]{\mathbb{E}[|\boldsymbol{x}_0-\mu|\boldsymbol{x}_t|^p]}$, where $\mu=\mathbb{E}[\boldsymbol{x}_0|\boldsymbol{x}_t]$. If the constraints function $\mathcal{R}$ approach $\mathcal{R}(\mu)$ no slower than $|\boldsymbol{x}_0-\mu|^\eta$ and grow as $\boldsymbol{x}_0\rightarrow\pm\infty$ no faster than $\pm|\boldsymbol{x}_0|^n$ for $n\geq\eta$, then the Jensen's gap $\mathbb{E}[\mathcal{R}\left(\boldsymbol{x}_0\right)\mid\boldsymbol{x}_t]-\mathcal{R}\left(\mathbb{E}[\boldsymbol{x}_0\mid\boldsymbol{x}_t]\right)$ approaches to 0 no slower than $\sigma_n^\eta$ as $\sigma_n\rightarrow 0$ (Gao et al., 2017). Usually, in the reverse diffusion process, $\sigma_n\rightarrow 0$ as $t\rightarrow 0$ since the generated noisy samples converge to a clean one. In this case, we use the penalty loss of $\mathcal{J}_{\mathcal{R}}\left(\boldsymbol{\theta}\right)=\mathbb{E}_{t,\boldsymbol{x}_0,\boldsymbol{\epsilon}}\left[w(t)\|\mathcal{R}\left(\boldsymbol{x}_{\boldsymbol{\theta}}\left(\boldsymbol{x}_t,t\right)\right)\|^2\right]$.

In the aforementioned three scenarios, at the implementation level, the model's output may be directly considered as the ground-truth sample $\boldsymbol{x}_0$ itself, rather than the conditional expectation $\mathbb{E}[\boldsymbol{x}_0\mid\boldsymbol{x}_t]$. These scenarios are referred to as "elementary cases". In the following, we will discuss how to deal with nonlinear cases using the above elementary cases.

**Reducible nonlinear cases.** For nonlinear constraints, mathematically speaking, we cannot directly apply the constraints to $\mathbb{E}[\boldsymbol{x}_0\mid\boldsymbol{x}_t]$. However, we may recursively use multilinear functions to decompose the nonlinear constraints into elementary ones as: $\mathcal{R}\left(\boldsymbol{x}_0\right)=\mathbf{g}_1\circ\cdots\mathbf{g}_m\left(\boldsymbol{x}_0\right)=\mathbf{0}$, where all $\mathbf{g}_i$ are elementary. Using elementary functions for decomposition, we may 1) reduce nonlinear constraints into elementary ones by treating terms causing nonlinearity as constants, and 2) reduce the complex constraints into several simpler ones. In this case, the penalty loss is set to $\mathcal{J}_{\mathcal{R}}\left(\boldsymbol{\theta}\right)=\mathbb{E}_{t,\boldsymbol{x}_0,\boldsymbol{\epsilon}}\left[w(t)\|\mathbf{g}_1\circ\cdots\mathbf{g}_m\left(\boldsymbol{x}_{\boldsymbol{\theta}}\left(\boldsymbol{x}_t,t\right)\right)\|^2\right]$. See Sec. 4.2 for concrete examples of nonlinear formulas for the conservation of energy.

**General nonlinear cases.** For general nonlinear cases, if it is not feasible to decompose the nonlinear constraints into their elementary components, it may be necessary to consider alternative approaches where we may reparameterize the constraints variable into elementary cases. Given the nonlinear constraints, we reparameterize it as $\mathcal{R}\left(\boldsymbol{x}_0\right)=\mathbf{g}\left(\mathbf{h}(\boldsymbol{x}_0)\right)=\mathbf{0}$, where $\mathbf{g}$ is elementary and $\mathbf{h}$ is non-necessarily elementary functions. Subsequently, another diffusion model, denoted as $\tilde{\boldsymbol{x}}_{\boldsymbol{\theta}}\left(\boldsymbol{x}_t,t\right)$, is trained to predict $\mathbf{h}(\boldsymbol{x}_0)$, utilizing the same hidden states as model $\boldsymbol{x}_{\boldsymbol{\theta}}\left(\boldsymbol{x}_t,t\right)$. This training process employs the methods applicable to elementary cases. The objective is for model $\tilde{\boldsymbol{x}}_{\boldsymbol{\theta}}$ to learn the underlying physical constraints and encode these constraints into its hidden states. Consequently, when model $\boldsymbol{x}_{\boldsymbol{\theta}}$ predicts, it inherently incorporates the learned physical constraints $\mathbf{g}$ parameterized by $\mathbf{h}(\boldsymbol{x}_0)$. To train model $\tilde{\boldsymbol{x}}_{\boldsymbol{\theta}}$, we set the penalty loss to be $\mathcal{J}_{\mathcal{R}}\left(\boldsymbol{\theta}\right)=\mathbb{E}_{t,\boldsymbol{x}_0,\boldsymbol{\epsilon}}\left[w(t)\|\tilde{\boldsymbol{x}}_{\boldsymbol{\theta}}\left(\boldsymbol{x}_t,t\right)-\mathbf{h}(\boldsymbol{x}_0)\|^2\right]$. See Appendix E.1 for implementation details.

Notably, in our proposed methods for integrating constraints, the explicit form of prior knowledge, such as the physics constants required for energy calculations, is not necessary. Instead, it suffices to determine whether the model's output parameters are elementary w.r.t. the constraints. This approach enhances the applicability of our methods to a broader spectrum of constraints.

**Remark 2.** *In conclusion, incorporating the physics constraints can be achieved in different ways depending on their complexity. For elementary constraints, one can directly omit Jensen's gap and impose the penalty loss on the model's output. In the case of nonlinear constraints, decomposition or reparameterization techniques are utilized to transform constraints into elementary ones.*

## 4 EXPERIMENTS

In this section, we assess the enhancement achieved by incorporating physics constraints into the fundamental diffusion model across various synthetic physics datasets. We conduct a grid search to identify an equivalent set of suitable hyperparameters for the network to perform the data/noise matching, ensuring a fair comparison between the baseline method (diffusion objectives without penalty loss) and our proposed approach of incorporating physics constraints. Appendix E provides

a detailed account of the selection of backbones and the training strategies employed for each dataset. We also provide ablation studies in Sec. 4.3 of **1**) data matching and noise matching techniques for different datasets, revealing that incorporating a distributional prior enhances model performance; **2**) the effect of omitting Jensen's gap, finding that nonlinear constraints can hinder performance if not properly handled. However, appropriately managing these priors using our proposed methods can lead to significant performance improvements.

## 4.1 PDE DATASETS

PDE datasets, including advection (Zang, 1991), Darcy flow (Li et al., 2024), Burgers (Rudy et al., 2017), and shallow water (Klöwer et al., 2018), are fundamental resources for studying and modeling various physical phenomena. These datasets enable the simulation of complex systems, demonstrating the capability of models for broader application across a wide range of PDE datasets. Through this, they facilitate advances in understanding diverse natural and engineered processes.

**Experiment settings.** The PDE constraints for the above datasets are given by:

$$\text{Advection:} \quad \partial_t u(t, x) + \beta \partial_x u(t, x) \qquad = 0, \tag{8a}$$

$$\text{Darcy flow:} \quad \partial_t u(x, t) - \nabla(a(x)\nabla u(x, t)) = f(x), \tag{8b}$$

$$\text{Burger:} \quad \partial_t u(x, t) + u(x, t)\partial_x u(x, t) \quad = 0, \tag{8c}$$

$$\text{Shallow water:} \quad \partial_t u = -\partial_x h, \quad \partial v_t = -\partial h_y, \quad \partial_t h = -c^2 \left( \partial_x u + \partial_y v \right). \tag{8d}$$

A detailed introduction and visualization of the datasets can be found in Appendix C.1. In this study, we investigate the predictive capabilities of generative models applied to advection and Darcy flow datasets. Our experiments focus on evaluating the models' accuracy in forecasting future states given initial conditions. Additionally, we examine the models' ability to generate physically feasible samples that align with the distribution of the training set on advection, Burger, and shallow water datasets. The evaluation metrics are designed to assess to what extent the solutions adhere to the physical feasibility constraints imposed by the corresponding PDEs.

**Injecting physical feasibility priors.** We train the models that apply the data matching objective as suggested in Remark 1. We employ finite difference methods to approximate the differential equations. This approach renders the PDE constraints linear for the advection, Darcy flow, and shallow water datasets. However, PDE constraints become multilinear for the Burgers' equation dataset (see Appendix C.2 for the proof). Thus, the first set of datasets: advection, Darcy flow, and shallow water—correspond to the linear case, while the Burgers' equation dataset corresponds to the multilinear case. We can directly apply the physical feasibility constraints on the model's output.

**Experimental results.** Results can be seen in Tab. 1, 2. In Tab. 1, we analyze the performance of diffusion models in predicting physical dynamics, given initial conditions, within a generative framework that produces a Dirac distribution. The accuracy of these models is evaluated using the RMSE metric. The observed loss magnitude is comparable to the prediction loss using with FNO, U-Net, and PINN models (Takamoto et al., 2022) (refer to Appendix E.4 for further details). Our results indicate that the incorporation of constraints consistently enhances the accuracy of the prediction. In Tab. 2, the feasibility of the generated samples is evaluated by calculating the RMSE of the PDE constraints, which determine the impact of incorporating physical feasibility priors on diffusion models. We also provide visualization of the generated samples in Fig. 11, 12, 13.

Table 1: Performance comparison of diffusion models with/without priors for predicting physical dynamics. The models' accuracy is measured using the RMSE metric, highlighting the impact of incorporating constraints on improving prediction accuracy.

| Method | Advection ($\times 10^{-2}$) | Darcy flow ($\times 10^{-2}$) |
|---|---|---|
| w/o prior | 1.7263±0.0491 | 2.0648±0.0600 |
| w/ prior | **1.6536±0.0677** | **1.9678±0.0651** |

Table 2: Comparative analysis of diffusion models, assessing the feasibility of generated samples with/without physical feasibility priors. We evaluate the RMSE of PDE constraints, demonstrating the effect of physical feasibility priors on the adherence to PDEs.

| Method | Advection | Burger | Shallow water |
|---|---|---|---|
| w/o prior | 2.398±0.024 | 6.862±0.060 | 8.0153±0.0960 |
| w/ prior | **2.305±0.001** | **6.610±0.012** | **7.7618±0.0645** |

## 4.2 PARTICLE DYNAMICS DATASETS

We train diffusion models to simulate the dynamics of chaotic three-body systems in 3D (Zhou & Yu, 2023) and five-spring systems in 2D (Kuramoto, 1975; Kipf et al., 2018) (see Appenidx. D.1 for visualizations of datasets). In the case of the three-body, we unconditionally generate the positions and velocities of three particles, where gravitational interactions govern their dynamics. The stochastic nature of this dataset arises from the random distribution of the initial positions and velocities. In five-spring systems, each pair of particles has a probability 50% of being connected by a spring. The movements of the particles are influenced by the spring forces, which cause stretching or compression interactions. We conditionally generate the positions and velocities of the five particles based on their spring connectivity.

**Notations.** The features of the datasets are represented as $\mathbf{X}^{(0)} = [\mathbf{C}^{(0)} \ \mathbf{V}^{(0)}] \in \mathbb{R}^{L \times K \times 2D}$, where $\mathbf{C}^{(0)}, \mathbf{V}^{(0)} \in \mathbb{R}^{L \times K \times D}$. Here, the matrix $\mathbf{C}^{(0)}$ encapsulates the coordinate features, while $\mathbf{V}^{(0)}$ encapsulates the velocity features. The superscript denotes the time for the diffusion process and the subscripts denote the matrix index. $L$ represents the temporal length of the physical dynamics, $K$ denotes the number of particles, and $D$ corresponds to the spatial dimensionality. We use the subscript $l$ to indicate time, while the subscripts $i, j$, and $k$ are used to denote the indices of particles. The subscript $d$ represents the index corresponding to the spatial axis. We also use the subscript of $\boldsymbol{\theta}$ to denote the corresponding values of the model's prediction of $\mathbb{E}[\mathbf{X}^{(0)} \mid \mathbf{X}^{(t)}]$ with inputs $\mathbf{X}^{(t)}$ and $t$, and $\mathbf{X}_{\boldsymbol{\theta}} = [\mathbf{C}_{\boldsymbol{\theta}} \ \mathbf{V}_{\boldsymbol{\theta}}]$.

**Injecting $\mathrm{SE(n)}$-invariance and permutation invariance.** Two physical dynamic systems are governed by the interactions between each pair of particles, resulting in a distribution that is $\mathrm{SE(n)}$ and permutation invariant. Our objective is to develop models that are $\mathrm{SO(n)}$-equivariant, translation invariant, and permutation equivariant. We intend to apply a noise matching objective to achieve the desired invariant distribution. However, to the best of our knowledge, no such architecture satisfying the above properties has been established within the context of diffusion generative models. Therefore, we opt to utilize a data augmentation method to ensure the model's equivariance and invariance properties (Chen et al., 2019; Botev et al., 2022), i.e. we apply these group operations in the training process, which enforces models to be equivariant and invariant.

**Conservation of momentum.** For both datasets, the momentum conservation is given by:

$$\sum_{k=1}^{K} m_k \mathbf{V}_{l,k,d}^{(0)} = \text{constant}_d, \quad \forall l = 1, \dots, L, d = 1, \dots, D. \tag{9}$$

Here, $m_k$ represents the mass of $k$-th particle, and $\mathbf{V}_{l,k,d}^{(0)}$ denotes the velocity along axis $d$ of the $k$-th particle at time $l$. The total momentum in each axis remains constant, as indicated by the equality. This constraint is linear w.r.t. $\mathbf{V}_{l,k,d}^{(0)}$, corresponding to the linear case. Let $\mathbf{f} : \mathbb{R}^{L \times K \times D} \times \mathbb{R}^K \to \mathbb{R}^D$ calculate the mean of the total momentum over time and set the penalty loss as

$$\mathcal{J}_{\mathcal{R}}(\boldsymbol{\theta}) = \mathbb{E}_{t,\boldsymbol{x}_0,\boldsymbol{\epsilon}} \left[ w(t) \sum_{l=1}^{L} \sum_{d=1}^{D} \left\| \left( \sum_{k=1}^{K} m_k (\mathbf{V}_{\boldsymbol{\theta}})_{l,k,d} \right) - \mathbf{f}_d(\mathbf{V}_{\boldsymbol{\theta}}, \{m_k\}_{k=1}^{K}) \right\|^2 \right]. \tag{10}$$

**Conservation of energy for the three-body dataset.** The total of gravitational potential energy and kinetic energy remains constant over time. The energy conservation equation is given by:

$$-\sum_{i \neq j}^{K} \frac{G m_i m_j}{\mathbf{R}_{l,ij}^{(0)}} + \sum_{k=1}^{K} \sum_{d=1}^{D} \frac{1}{2} m_k (\mathbf{V}_{l,k,d}^{(0)})^2 = \text{constant}, \quad \forall l = 1, \dots, L, \tag{11}$$

where $G$ denotes the gravitational constant. $\mathbf{R}_{l,ij}^{(0)} = \|\mathbf{C}_{l,i}^{(0)} - \mathbf{C}_{l,j}^{(0)}\|$ denotes the Euclidean distance between the $i$-th and $j$-th particle at time $l$. This constraint is nonlinear with $\mathbf{X}^{(0)}$ but can be decomposed into elementary cases. Note that the constraint is multilinear w.r.t. $1/\mathbf{R}_{l,ij}^{(0)}$ and $(\mathbf{V}_{l,k,d}^{(0)})^2$. Hence, from the results of the general nonlinear cases, we can train another model sharing the same

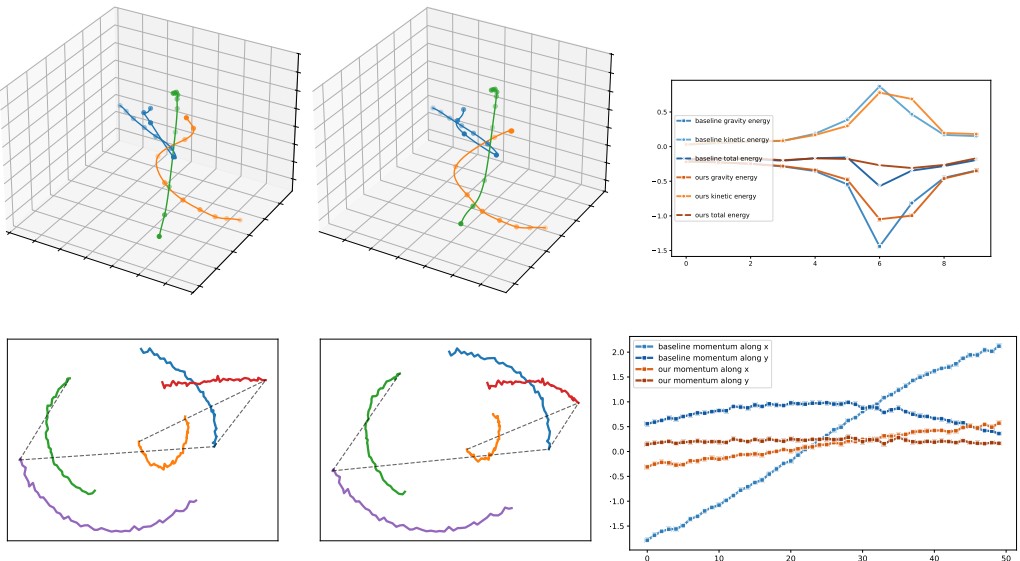

Figure 2: Visualization of generated samples from the three-body (first row) and five-spring (second row) datasets. The leftmost figures in each row represent methods without priors, the middle figures correspond to our proposed methods, and the rightmost figures illustrate the physical properties as they evolve over time. Both total momentum and total energy should remain conserved. The samples generated by our methods demonstrate stronger adherence to physical feasibility.

hidden size as the model for noise matching to predict these variables related to the conservation of energy. Furthermore, since these variables are convex w.r.t. $\mathbf{X}^{(0)}$, by the results of the convex case, we can directly apply the penalty loss as:

$$\mathcal{J}_{\mathcal{R}}\left(\boldsymbol{\theta}\right) = \mathbb{E}_{t,\boldsymbol{x}_0,\boldsymbol{\epsilon}}\left[w_1(t)\sum_{l;i\neq j}\left\|\frac{1}{(\mathbf{R}_{\boldsymbol{\theta}})_{l,ij}} - \frac{1}{\mathbf{R}_{l,ij}^{(0)}}\right\|^2 + w_2(t)\sum_{l,k,d}\left\|(\mathbf{V}_{\boldsymbol{\theta}})_{l,k,d}^2 - \left(\mathbf{V}_{l,k,d}^{(0)}\right)^2\right\|^2\right], \quad (12)$$

where $(\mathbf{R}_{\boldsymbol{\theta}})_{l,ij} = \|\mathbf{C}_{\boldsymbol{\theta}}\left(\mathbf{X}^{(t)},t\right)_{l,i} - \mathbf{C}_{\boldsymbol{\theta}}\left(\mathbf{X}^{(t)},t\right)_{l,j}\|$, i.e. model's prediction of the Euclidean distance between two particles calculated from its prediction of coordinates. This penalty loss can also be derived from the reducible case. The detailed derivation can be found in Appendix D.2.

**Conservation of energy for the five-spring dataset.** The combined elastic potential energy and the kinetic energy are conserved throughout time. The equation for the conservation of energy is represented by:

$$\sum_{(i,j)\in\mathcal{E}}\frac{1}{2}\kappa\left(\mathbf{R}_{l,ij}^{(0)}\right)^2 + \sum_{k=1}^{K}\sum_{d=1}^{D}\frac{1}{2}m_k\left(\mathbf{V}_{l,k,d}^{(0)}\right)^2 = \text{constant}, \quad \forall l = 1,\ldots,L, \quad (13)$$

where $\kappa$ denotes the elastic constant, $\mathbf{R}_{l,ij}^{(0)} = \|\mathbf{C}_{l,i}^{(0)} - \mathbf{C}_{l,j}^{(0)}\|$ denotes the distance between the $i$-th and $j$-th particle at time $l$, and $\mathcal{E}$ denotes the edge set of springs connecting particles. $m_k$ represents the mass of the $k$-th particle. Analogue to the conservation of energy for the three-body dataset, we can reduce the nonlinear constraints into elementary cases.

**Experimental results.** The results can be seen in Tab. 3 and Fig. 2, 14, 15, and we refer readers to Appendix E.2 for a detailed account of the experimental settings, as well as a more extensive comparison of the effects of hyperparameters and various methods for injecting constraints across the discussed cases. Our analysis indicates that for the three-body dataset, the incorporation of the conservation of energy prior, via the reducible case method, substantially enhances the model's performance across all evaluated metrics. Similarly, applying the conservation of momentum prior to the five-spring dataset significantly reduces the momentum error in the generated samples. This also

Table 3: Sample quality of the three-body and five-spring datasets. For both datasets, we simulate the ground-truth future motion based on the current states of the generated samples and report the MSE error between the ground-truth motion and the generated ones. We also calculate the error of physical feasibility such as conservation of the energy and momentum, which should remain unchanged along the evolution of the systems.

| Method | Three-body | | | Five-spring | | |
|---|---|---|---|---|---|---|
| | Traj error | Vel error | Energy error | Dynamic error | Momentum error | Energy error |
| w/o prior | 2.4132±0.1208 | 2.5745±0.0790 | 4.3292±0.7235 | 5.1754±0.0286 | 5.3699±0.0462 | 1.0618±0.0243 |
| w/ prior | **1.9880±0.3418** | **0.8328±0.1042** | **0.5465±0.0705** | **5.0731±0.0406** | **0.3898±0.0118** | **0.7418±0.0129** |

contributes to a reduction in the errors associated with dynamics and energies. Fig. 2 demonstrates that the total momentum and energy of samples generated with the incorporation of priors exhibit greater stability compared to those without priors. We also provide sampling results using the DPM-solvers (Lu et al., 2022) in Appendix E.7, which significantly lower computational expenses.

## 4.3 ABLATION STUDIES

**Distributional priors through matching objective.** We employ data matching and noise matching techniques for the PDE and particle dynamics datasets, respectively. An ablation study is conducted to investigate the effects of applying the alternative matching objective on the particle dynamics and PDE datasets, both without physical feasibility priors. The results, presented in Tab. 4, demonstrate that incorporating a distributional prior can significantly improve the model's performance.

**Omitting Jensen's gap.** In the three-body dataset, we employ a multilinear function to simplify constraints into convex scenarios. We now conduct an ablation study in which the output of a diffusion model is considered the ground-truth, and the constraint of energy conservation

Table 4: Results of an ablation study comparing the effects of data matching and noise matching techniques. The findings show that incorporating a distributional prior improves model performance. We use the mean of trajectory error and velocity error as the metric for the three-body dataset.

| Method | Three-body | Five-spring | Darcy flow | Shallow water |
|---|---|---|---|---|
| distributional prior | **2.6084** | **5.1929** | **2.016** | **8.150** |
| alternative | 4.7241 | 5.3120 | 7.268 | 27.40 |

Table 5: Results show the impact of enforcing energy conservation constraints on the three-body dataset. Direct application of nonlinear constraints (prior by PINN) can degrade performance, while proper handling (prior by ours) improves accuracy.

| Method | Traj error | Vel error | Energy error |
|---|---|---|---|
| w/o prior | 2.5613 | 2.6555 | 3.8941 |
| prior by PINN | 2.6048 | 2.6437 | 4.2219 |
| prior by ours | **1.6072** | **0.7307** | **0.5062** |

is imposed similarly to the injection of constraints by penalty loss in the prediction tasks of PINNs. This configuration is referred to as "prior by PINN". We define a penalty loss based on the variation of energy over time, analogous to the penalty loss used to enforce momentum conservation constraints. However, unlike the conservation of momentum, which is governed by a linear constraint and can thus be applied directly, the conservation of energy involves a nonlinear constraint. This introduces Jensen's gap, preventing the direct application of the constraint. The results, presented in Tab. 5, indicate that directly applying nonlinear constraints can degrade the model's performance. However, appropriately handling these constraints can significantly improve the sample quality.

## 5 CONCLUSION

In conclusion, this paper presents a novel method for generating physically feasible dynamics using diffusion models by integrating distributional and physical feasibility priors. We inject distributional priors through equivariant models and noising matching, while incorporating physical feasibility priors through constraint decomposition. Empirical results demonstrate the robustness of our method across various physical phenomena, highlighting its promise for data-driven AI4Physics research. This work emphasizes the importance of embedding domain knowledge into learning systems, bridging physics and machine learning through innovative use of physical priors.

ACKNOWLEDGMENTS

This work was supported by the National Science and Technology Major Project of China under Grant 2022ZD0116408. This work is also supported by a grant from ChemLex Technology Co., Ltd..

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

# A RELATED WORK

## A.1 GENERATIVE METHODS FOR PHYSICS

Numerous studies have been conducted on the development of surrogate models to supplant numerical solvers for physics dynamics with GANs (Farimani et al., 2017; de Oliveira et al., 2017; Wu et al., 2020; Yang et al., 2020; Bode et al., 2021) and VAEs (Cang et al., 2018). Nevertheless, to generate realistic physics dynamics, one must accurately learn the data distribution or inject physics prior (Cuomo et al., 2022). Recent advancements in diffusion models (Song et al., 2020) have sparked increased interest in their direct application to the generation and prediction of physical dynamics, circumventing the need for specific physics-based formulations (Shu et al., 2023; Lienen et al., 2023; Yang & Sommer, 2023; Apte et al., 2023; Jadhav et al., 2023; Bastek et al., 2024). However, these approaches, which do not incorporate prior physical knowledge, may exhibit limited performance, potentially leading to suboptimal results.

## A.2 SCORE-BASED DIFFUSION MODELS

Score-based diffusion models are a class of generative models that create high-quality data samples by progressively refining noise into detailed data through a series of steps (Song et al., 2020). These models estimate the score function, the gradient of the log-probability density of the data, using a neural network (Song & Ermon, 2019). By applying this score function iteratively to noisy samples, the model reverses the diffusion process, effectively denoising the data incrementally, and generates samples following the same distribution as the training set. Although numerous studies on diffusion models have focused on generating SE(3)-invariant distributions (Xu et al., 2022; Yim et al., 2023; Zhou et al., 2024), there remains a lack of comprehensive research on the generation of general invariant distributions under group operations. Meanwhile, in contrast to GANs, the outputs produced by diffusion models represent the distributional properties of the data samples. This fundamental difference means that physical feasibility priors cannot be added directly to the model output due to the presence of a Jensen gap (Chung et al., 2022), i.e. $\mathcal{R}\left(\mathbb{E}[\boldsymbol{x}_0 \mid \boldsymbol{x}_t]\right) \neq \mathbb{E}[\mathcal{R}\left(\boldsymbol{x}_0\right) \mid \boldsymbol{x}_t]$. A potential solution to this problem involves iterating and drawing samples during the training process and subsequently incorporating the loss of physics feasibility on the generated samples (Bastek et al., 2024). However, this approach necessitates numerous iterations, often in the hundreds, rendering the training process inefficient.

## A.3 PHYSICS-INFORMED NEURAL NETWORKS

Physics-Informed Neural Networks (PINNs) are a class of deep learning models that incorporate physical laws and constraints into their training process (Lawal et al., 2022). Unlike traditional training processes, which learn patterns solely from data, PINNs leverage priors including PDEs that describe physical phenomena to guide the learning process. By incorporating these physical feasibility equations as part of the penalty loss, alongside the data prediction loss, PINNs enhance their ability to model complex systems. This integration allows PINNs to be applied across various fields, including fluid dynamics (Cai et al., 2021), electromagnetism (Khan & Lowther, 2022), and climate modeling (Hwang et al., 2021). Their ability to integrate domain knowledge with data-driven learning makes them a powerful tool for tackling complex scientific and engineering challenges.

# B EXTENSION ON EQUIVALENCE CLASS MANIFOLD

## B.1 FORMAL DEFINITION

Let $X$ be a set and $\sim$ be an equivalence relation on $X$. The equivalence class manifold $\mathcal{M}$ is defined as the set of equivalence classes under the relation $\sim$. Formally,

$$\mathcal{M} = \{x \mid x \in X, y \in [x] \Rightarrow y = x\}, \tag{14}$$

where $\mathcal{M}$ is a Riemannian manifold and $[x]$ denotes the equivalence class of $x$, defined as:

$$[x] = \{y \in X \mid y \sim x\}. \tag{15}$$

## B.2 EQUIVALENCE CLASS MANIFOLD OF SE(3)-INVARIANT DISTRIBUTION

The following theorem provides a method to use a set of coordinates to represent all other coordinates having the same pairwise distances.

**Theorem 4** (Equivalence class manifold of SE(3)-invariant distribution (Dokmanic et al., 2015; Hoffmann & Noé, 2019; Zhou et al., 2024)). *Given any pairwise distance matrix $D \in \mathbb{R}_+^{n \times n}$, there exists a corresponding Gram matrix $M \in \mathbb{R}^{n \times n}$ defined by*

$$M_{ij} = \frac{1}{2}(D_{1j} + D_{i1} - D_{ij}) \tag{16}$$

*and conversely*

$$D_{ij} = M_{ii} + M_{jj} - 2M_{ij}. \tag{17}$$

*By performing the singular value decomposition (SVD) on the Gram matrix $M \in \mathbb{R}^{n \times n}$ (associated with $D$), we obtain exactly three positive eigenvalues $\lambda_1, \lambda_2, \lambda_3$ and their respective eigenvectors $\mathbf{v}_1, \mathbf{v}_2, \mathbf{v}_3$, where $\lambda_1 \geq \lambda_2 \geq \lambda_3 > 0$. The set of coordinates*

$$\mathcal{C} = [\mathbf{v}_1, \mathbf{v}_2, \mathbf{v}_3] \begin{bmatrix} \sqrt{\lambda_1} & 0 & 0 \\ 0 & \sqrt{\lambda_2} & 0 \\ 0 & 0 & \sqrt{\lambda_3} \end{bmatrix} \tag{18}$$

*satisfies has the same pairwise distance matrix as $D$.*

Define the above mapping from the pairwise distances to coordinates as $f$. Then, the equivalence class manifold of SE(3)-invariant distribution can be given by

$$\mathcal{M} = \{f(D) \mid D \text{ is a pairwise distance matrix}\}. \tag{19}$$

$\mathcal{M}$ satisfies the property of being a Riemannian manifold (Zhou et al., 2024).

## C PDE DATASETS

A summary of the important properties of datasets can be found in Tab. 6.

Table 6: Comparative summary of datasets. The table highlights key aspects such as the type of generation (conditional or unconditional), the matching objective, the distributional priors, and the nature of the constraint cases.

| Datasets | | Cond/Uncond generation | Matching objective | Distributional priors | Constraint cases |
|---|---|---|---|---|---|
| PDE | advection | both | data (Equation 6b) | PDE constraints | linear |
| | Darcy flow | conditional | | | linear |
| | Burger | unconditional | | | multilinear |
| | shallow water | conditional | | | linear |
| particle dynamics | three-body | unconditional | noise (Equation 6a) | SE(3) + permutation invariant | all cases |
| | five-spring | conditional | | SE(2) + permutation invariant | all cases |

## C.1 DATASET SETTINGS

**Advection.** The advection equation is a fundamental model in fluid dynamics, representing the transport of a scalar quantity by a velocity field. The dataset presented herein consists of numerical solutions to the linear advection equation, characterized by

$$\partial_t u(t, x) + \beta \partial_x u(t, x) = 0, \quad x \in (0, 1), t \in (0, 2] \tag{20}$$

where $u$ denotes the scalar field and $\beta = 0.1$ is a constant advection speed. The visualization of training samples can be seen in Fig. 3. Based on the initial conditions provided for the advection equation, our model utilizes a generative framework to predict the subsequent dynamics, with the specific aim of forecasting the next 40 frames. We then compare these predictions with the ground-truth to assess performance. Additionally, we evaluate the model's capability to generate samples unconditionally, without initial conditions, and measure performance using the physical feasibility implied by the PDE constraints.

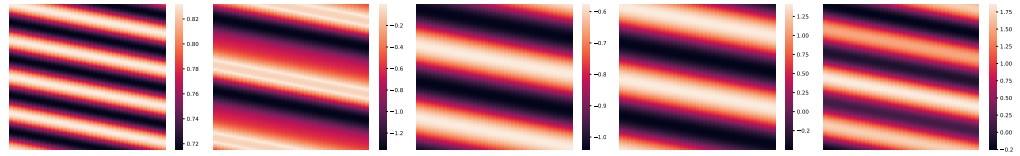

Figure 3: Samples from the advection dataset with varying initial conditions. The horizontal axis represents the spatial coordinate $x$, while the vertical axis represents the parameter $t$.

**Darcy flow.** Darcy's law describes the flow of fluid through a porous medium, which is a fundamental principle in hydrogeology, petroleum engineering, and other fields involving subsurface flow. The mathematical formulation of the Darcy flow PDE is given by:

$$\partial_t u(x, t) - \nabla(a(x)\nabla u(x, t)) = f(x), \quad x \in (0, 1)^2, \tag{21}$$

where $u(x, t)$ represents the fluid pressure at location $x$ and time $t$, $a(x)$ is the permeability or hydraulic conductivity, and $f(x)$ denotes sources or sinks within the medium. Given the initial state at $t = 0$, we use the generative scheme to forecast the state at $t = 1$. Fig. 4 provides a visualization of training samples. The accuracy of these predictions is evaluated by comparing them with the ground-truth values.

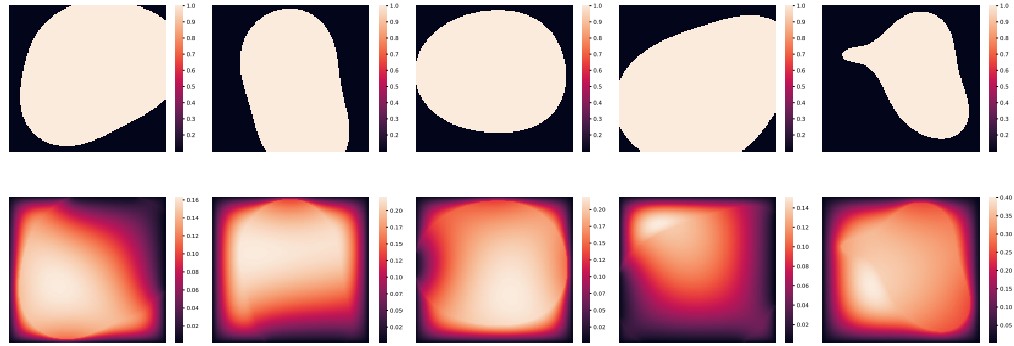

Figure 4: The figure illustrates representative training samples from the Darcy flow dataset. The first row displays the values for the function $a(x)$, while the second row shows the values of $u(x, t)$ at time $t = 1$.

**Burger.** The Burgers' equation is a fundamental PDE that appears in various fields such as fluid mechanics, nonlinear acoustics, and traffic flow. It is a simplified model that captures essential features of convection and diffusion processes. The equation is given by:

$$\partial_t u(x, t) + u(x, t)\partial_x u(x, t) = 0, \tag{22}$$

where $u(x, t)$ represents the velocity field, $x$ and $t$ denote spatial and temporal coordinates, respectively. We unconditionally generate samples following the distribution of the training set and evaluate feasibility within the realm of physics as dictated by the constraints of PDE.

**Shallow water.** The linearized 2D shallow water equations describe the dynamics of fluid flows under the assumption that the horizontal scale is significantly larger than the vertical depth. These equations are instrumental in fields such as oceanography, meteorology, and hydrology for modeling wave and current phenomena in shallow water regions. Let $u$ and $v$ denote the components of the velocity field in the $x$- and $y$-directions, respectively. The variable $h$ represents the perturbation in the free surface height of the fluid from a mean reference level. The parameter $c$ denotes the phase speed of shallow water waves, which is a function of the gravitational acceleration and the mean water depth. The equations are expressed as follows:

$$\frac{\partial u}{\partial t} = -\frac{\partial h}{\partial x}, \quad \frac{\partial v}{\partial t} = -\frac{\partial h}{\partial y}, \quad \frac{\partial h}{\partial t} = -c^2 \left( \frac{\partial u}{\partial x} + \frac{\partial v}{\partial y} \right). \tag{23}$$

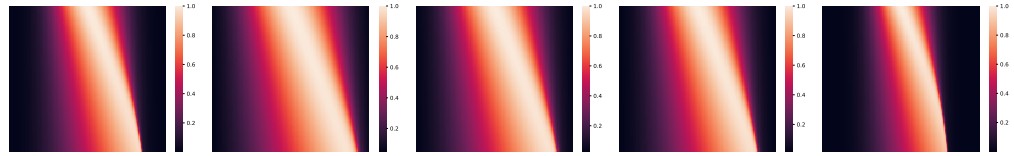

Figure 5: The figure depicts representative training samples from the Burger dataset. The samples within this dataset exhibit minimal variability. The horizontal axis denotes the spatial coordinate $x$, whereas the vertical axis represents the parameter $t$.

We conditionally generate the dynamics of shallow water expressed by $h, u, v$ conditioned on the given $c$. We provide a visualization of one sample in Fig. 6.

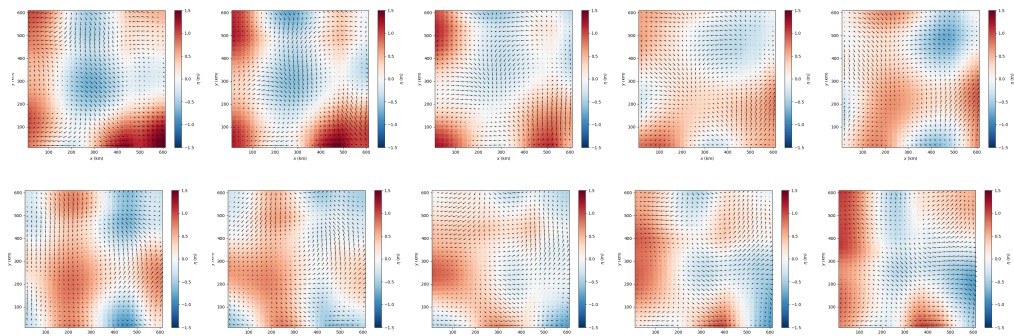

Figure 6: In the figure, the sequence from left to right represents a sample of the dynamics of shallow water. Each sample consists of 50 frames, from which 10 frames have been uniformly selected for visualization.

### C.2 CONVERTING TO ELEMENTARY CASES BY FINITE DIFFERENCE APPROXIMATION

**Advection and shallow water.** The original constraint of the advection equation is given by

$$\partial_t u(t, x) + \beta \partial_x u(t, x) = 0. \tag{24}$$

If we use the finite difference method to approximate the derivative, assume a grid with time steps $t_n$ and spatial points $x_i$. Let $u_n^i$ denote the approximation to $u(t_n, x_i)$. For the time derivative, use a forward difference approximation: $\partial_t u \approx \frac{u_{n+1}^i - u_n^i}{\Delta t}$. For the spatial derivative, use a central difference approximation: $\partial_x u \approx \frac{u_n^{i+1} - u_n^i}{\Delta x}$. Substituting these approximations into the PDE, we have

$$\frac{u_{n+1}^i - u_n^i}{\Delta t} + \beta \frac{u_n^{i+1} - u_n^i}{\Delta x} = 0. \tag{25}$$

Rearrange to obtain an equation that involves only $u$ values and constants:

$$u_{n+1}^i - (1 + \beta \frac{\Delta t}{\Delta x})u_n^i + \beta \frac{\Delta t}{\Delta x} u_n^{i+1} = 0. \tag{26}$$

In this form, the constraint is a linear equation involving $u_{n+1}^i, u_n^i, u_n^{i+1}$. The linearization of the shallow water constraints can be performed in an analogous manner.

**Darcy flow.** The given Darcy flow equation is:

$$\partial_t u(x, t) - \nabla(a(x)\nabla u(x, t)) = f(x), \tag{27}$$

Using the finite difference method, we discretize the domain into a grid with grid spacing $\Delta x = \Delta y$ and $\Delta t$. Let $u_{i,j}$ represent $u(x_i, y_j, t_n)$ and $u^n$ represent $u(x_i, y_j, t_n)$. The finite difference

approximations for the gradients and divergence are:

$$\partial_t u \approx \frac{u^{n+1} - u^{n-1}}{2\Delta t}, \tag{28a}$$

$$\partial_x u \approx \frac{u_{i+1,j} - u_{i-1,j}}{2\Delta x}, \tag{28b}$$

$$\partial_y u \approx \frac{u_{i,j+1} - u_{i,j-1}}{2\Delta y}. \tag{28c}$$

The Hessian matrix of $\nabla^2 u(x,t)$ is also linear w.r.t. $u(x,t)$ when approximated by the finite difference method. Hence, the left-hand side of the Darcy flow equation is the sum of terms linear w.r.t. $u(x,t)$ and thus the constraint is linear.

**Burger.** The partial differential equation

$$\partial_t u(x,t) + u(x,t)\partial_x u(x,t) = 0 \tag{29}$$

can be approximated using finite differences as follows: time derivative (forward difference): $\partial_t u(x,t) \approx \frac{u(x,t+\Delta t) - u(x,t)}{\Delta t}$, spatial derivative (central difference): $\partial_x u(x,t) \approx \frac{u(x+\Delta x,t) - u(x-\Delta x,t)}{2\Delta x}$. Substituting these into the PDE gives:

$$\frac{u(x,t+\Delta t) - u(x,t)}{\Delta t} + u(x,t) \cdot \frac{u(x+\Delta x,t) - u(x-\Delta x,t)}{2\Delta x} = 0 \tag{30}$$

Hence, the constraint is multilinear w.r.t. $u(x,t)$ if we consider values of $u$ at other points as constants.

## D  PARTICLE DYNAMICS DATASETS

### D.1  DATASET INTRODUCTION

The dataset features are structured as $\mathbf{X}^{(0)} = [\mathbf{C}^{(0)}\ \mathbf{V}^{(0)}]$, where $\mathbf{C}^{(0)}$ and $\mathbf{V}^{(0)}$ are both elements of $\mathbb{R}^{L \times K \times D}$. In this context, $L$ refers to the temporal length of the physical dynamics, $K$ represents the number of particles, and $D$ denotes the spatial dimensionality. Specifically, $\mathbf{C}^{(0)}$ captures the coordinate features, and $\mathbf{V}^{(0)}$ captures the velocity features. For the three-body dataset, the parameters are set as $L = 10$, $K = 3$, and $D = 3$, indicating a temporal length of 10, with 3 particles in a 3-dimensional space. Similarly, for the five-spring dataset, $L = 50$, $K = 5$, and $D = 2$, corresponding to a temporal length of 50, 5 particles, and a 2-dimensional space. For both datasets, we generated 50k samples for training. We aim to generate samples following the same distribution as in the training dataset.

Fig. 7 provides visual representations of two samples from the particle dynamics dataset, showcasing the behavior of systems within the three-body and five-spring datasets.

### D.2  DETAILS OF INJECTING THE CONSERVATION OF ENERGY FOR THE THREE-BODY DATASET

The total of gravitational potential energy (GPE) and kinetic energy (KE) remains constant over time. The formula of the energy conservation equation is given by:

$$-\sum_{i \neq j}^{K} \frac{Gm_i m_j}{\mathbf{R}_{l,ij}^{(0)}} + \sum_{k=1}^{K} \sum_{d=1}^{D} \frac{1}{2} m_k (\mathbf{V}_{l,k,d}^{(0)})^2 = \text{constant}, \quad \forall l = 1, \ldots, L, \tag{31}$$

where $G$ denotes the gravitational constant, and all three bodies have the same mass $m_k$. $\mathbf{R}_{l,ij}^{(0)} = \|\mathbf{C}_{l,i}^{(0)} - \mathbf{C}_{l,j}^{(0)}\|$ denotes the Euclidean distance between the $i$-th and $j$-th mass at time $l$. This constraint is nonlinear with $\mathbf{X}^{(0)}$ but can be decomposed into elementary cases. Note that the constraint

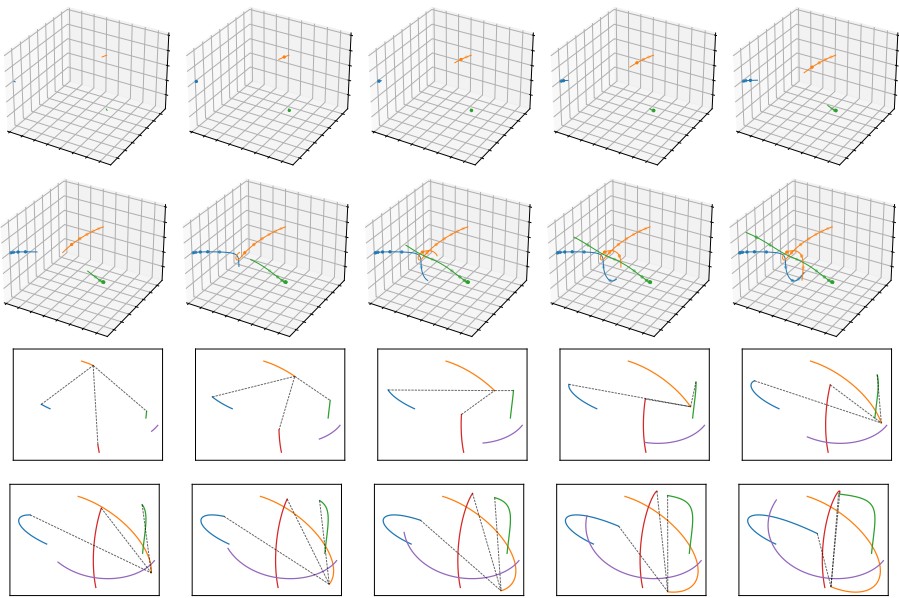

Figure 7: The presented figures illustrate samples from the particle dynamics dataset. The first two rows depict a sample from the three-body dataset, while the subsequent two rows represent a sample from the five-spring dataset.

is multilinear w.r.t. $1/\mathbf{R}^{(0)}_{l,ij}$ and $(\mathbf{V}^{(0)}_{l,k,d})^2$. Hence, from the results of the general nonlinear cases, we can apply the penalty loss $\mathcal{J}_{\mathcal{R}}(\boldsymbol{\theta}) = \mathcal{J}_{\text{GPE}}(\boldsymbol{\theta}) + \mathcal{J}_{\text{KE}}(\boldsymbol{\theta})$, and

$$\mathcal{J}_{\text{GPE}}(\boldsymbol{\theta}) = \mathbb{E}_{t,\boldsymbol{x}_0,\boldsymbol{\epsilon}}\left[w_1(t)\sum_{l;i\neq j}\left\|\left(\tilde{\mathbf{R}}_{\boldsymbol{\theta}}\right)_{l,ij} - \frac{1}{\mathbf{R}^{(0)}_{l,ij}}\right\|^2\right], \tag{32a}$$

$$\mathcal{J}_{\text{KE}}(\boldsymbol{\theta}) = \mathbb{E}_{t,\boldsymbol{x}_0,\boldsymbol{\epsilon}}\left[w_2(t)\sum_{l,k,d}\left\|\left(\tilde{\mathbf{V}}_{\boldsymbol{\theta}}\right)_{l,k,d} - \left(\mathbf{V}^{(0)}_{l,k,d}\right)^2\right\|^2\right], \tag{32b}$$

where $\tilde{\mathbf{R}}_{\boldsymbol{\theta}}$ and $\tilde{\mathbf{V}}_{\boldsymbol{\theta}}$ share the same hidden state as the model $\mathbf{X}_{\boldsymbol{\theta}}$ and $\mathbf{X}_{\boldsymbol{\theta}}$ predicts $\mathbb{E}[\mathbf{X}^{(0)} \mid \mathbf{X}^{(t)}]$. The setting details of these models are introduced in Appendix E.1. We refer such a setting as "*noise matching + conservation of energy (general nonlinear)*". Meanwhile, note that $1/\mathbf{R}^{(0)}_{l,ij}$ and $(\mathbf{V}^{(0)}_{l,k,d})^2$ are convex w.r.t. model's output. From the results in the convex case, we can directly apply the penalty loss to the output of $\mathbf{X}_{\boldsymbol{\theta}}$ and set the penalty loss to be

$$\mathcal{J}_{\text{GPE}}(\boldsymbol{\theta}) = \mathbb{E}_{t,\boldsymbol{x}_0,\boldsymbol{\epsilon}}\left[w_1(t)\sum_{l;i\neq j}\left\|\frac{1}{(\mathbf{R}_{\boldsymbol{\theta}})_{l,ij}} - \frac{1}{\mathbf{R}^{(0)}_{l,ij}}\right\|^2\right], \tag{33a}$$

$$\mathcal{J}_{\text{KE}}(\boldsymbol{\theta}) = \mathbb{E}_{t,\boldsymbol{x}_0,\boldsymbol{\epsilon}}\left[w_2(t)\sum_{l,k,d}\left\|(\mathbf{V}_{\boldsymbol{\theta}})^2_{l,k,d} - \left(\mathbf{V}^{(0)}_{l,k,d}\right)^2\right\|^2\right], \tag{33b}$$

where $(\mathbf{R}_{\boldsymbol{\theta}})_{l,ij} = \|\mathbf{C}_{\boldsymbol{\theta}}\left(\mathbf{X}^{(t)}, t\right)_{l,i} - \mathbf{C}_{\boldsymbol{\theta}}\left(\mathbf{X}^{(t)}, t\right)_{l,j}\|$, i.e. model's prediction of the Euclidean distance between two masses. We refer such a setting as "*noise matching + conservation of energy (reducible nonlinear)*", since this penalty loss function can be derived using a multilinear function composed with convex functions. In comparison to the penalty loss described in Equation 32, the penalty loss presented in Equation 33 is applied directly to the output of $\mathbf{X}_{\boldsymbol{\theta}}$. This direct application imposes a stronger constraint, thereby more effectively ensuring that the model adheres to the specified physical constraints.

## E  EXPERIMENTS DETAILS

**Configuration.**  We conduct experiments of advection, Darcy flow, three-body, and five-spring datasets on NVIDIA GeForce RTX 3090 GPUs and Intel(R) Xeon(R) Silver 4210R CPU @ 2.40GHz CPU. For the rest of the datasets, we conduct experiments on NVIDIA A100-SXM4-80GB GPUs and Intel(R) Xeon(R) Platinum 8358P CPU @ 2.60GHz CPU.

**Training details.**  We use the Adam optimizer for training, with a maximum of 1000 epochs. We set the learning rate to 1e-3 and betas to 0.95 and 0.999. The learning rate scheduler is ReduceLROnPlateau with factor=0.6 and patience=10. When the learning rate is less than 5e-7, we stop the training.

**Diffusion details.**  As for diffusion configuration, we set the steps of the forward diffusion process to 1000, the noise scheduler to $\sigma_t = \text{sigmoid}(\text{linspace}(-5, 5, 1000))$ and $\alpha_t = \sqrt{1 - \sigma_t^2}$. The loss weight $w(t)$ is set to $g^2(t) = \frac{\mathrm{d}\sigma_t^2}{\mathrm{d}t} - 2\frac{\mathrm{d}\log\alpha_t}{\mathrm{d}t}\sigma_t^2$ (Song et al., 2021). We generate samples using the DPM-Solver-1 (Lu et al., 2022).

**Experiment summary.**  We summarize the choice for backbones and properties of datasets in Tab. 6. We conducted an equivalent search for the hyperparameters of both the baseline methods and the proposed methods. The specific search ranges for each dataset and the corresponding hyperparameters are summarized in Tab. 7.

Table 7: Summary of the model hyperparameters.

| Datasets | | Backbone | Model hyperparameters | Batch size |
|---|---|---|---|---|
| PDE | advection | GRU (Chung et al., 2014) | hidden size: [128, 256, 512], layers: [3, 4, 5] | 128 |
| | Darcy flow | Karras Unet (Ho et al., 2020) | dim: [128] | 8 |
| | Burger | Karras Unet (Ho et al., 2020) | dim: [32] | 128 |
| | shallow water | 3D Karras Unet (Ho et al., 2020) | dim: [16] | 64 |
| particle dynamics | three-body | NN+GRU (Chung et al., 2014) | RNN hidden size: [64, 128, 256, 512, 1024], layers: [3, 4, 5] | 64 |
| | five-spring | EGNN (Satorras et al., 2021)+GRU (Chung et al., 2014) | RNN hidden size: [256, 512, 1024] | 64 |

**Error bars.**  Error bars are computed by varying both training and sampling seeds (3-5 runs), except for Darcy flow, where only the sampling seed is varied due to the long training time.

### E.1  TRAINING GENERAL NONLINEAR CASES

**Three-body dataset.**  To reduce the nonlinear conservation of the energy by general nonlinear cases, we apply the penalty loss $\mathcal{J}_{\mathcal{R}}(\boldsymbol{\theta}) = \mathcal{J}_{\text{GPE}}(\boldsymbol{\theta}) + \mathcal{J}_{\text{KE}}(\boldsymbol{\theta})$, and

$$\mathcal{J}_{\text{GPE}}(\boldsymbol{\theta}) = \mathbb{E}_{t,\boldsymbol{x}_0,\boldsymbol{\epsilon}}\left[w_1(t)\sum_{l;i\neq j}\left\|\left(\tilde{\mathbf{R}}_{\boldsymbol{\theta}}\right)_{l,ij} - \frac{1}{\mathbf{R}_{l,ij}^{(0)}}\right\|^2\right], \quad (34a)$$

$$\mathcal{J}_{\text{KE}}(\boldsymbol{\theta}) = \mathbb{E}_{t,\boldsymbol{x}_0,\boldsymbol{\epsilon}}\left[w_2(t)\sum_{l,k,d}\left\|\left(\tilde{\mathbf{V}}_{\boldsymbol{\theta}}\right)_{l,k,d} - \left(\mathbf{V}_{l,k,d}^{(0)}\right)^2\right\|^2\right]. \quad (34b)$$

The models $\tilde{\mathbf{R}}_{\boldsymbol{\theta}}$ and $\tilde{\mathbf{V}}_{\boldsymbol{\theta}}$ share the same hidden state as the model $\mathbf{X}_{\boldsymbol{\theta}}$, where $\mathbf{X}_{\boldsymbol{\theta}}$ is tasked with predicting $\mathbb{E}[\mathbf{X}^{(0)} \mid \mathbf{X}^{(t)}]$. The GRU architecture serves as the backbone for $\mathbf{X}_{\boldsymbol{\theta}}$. Consequently, we have designed the outputs of the models $\tilde{\mathbf{R}}_{\boldsymbol{\theta}}$ and $\tilde{\mathbf{V}}_{\boldsymbol{\theta}}$ to be generated by an additional linear layer that takes the hidden state of the GRU within $\mathbf{X}_{\boldsymbol{\theta}}$ as input.

**Five-spring dataset.** For the five-spring dataset, we apply the penalty loss $\mathcal{J}_{\mathcal{R}}(\boldsymbol{\theta}) = \mathcal{J}_{\mathrm{PE}}(\boldsymbol{\theta}) + \mathcal{J}_{\mathrm{KE}}(\boldsymbol{\theta})$, and

$$\mathcal{J}_{\mathrm{PE}}(\boldsymbol{\theta}) = \mathbb{E}_{t,\boldsymbol{x}_0,\boldsymbol{\epsilon}}\left[w_1(t)\sum_{(i,j)\in\mathcal{E},l}\left\|\left(\tilde{\mathbf{R}}_{\boldsymbol{\theta}}\right)_{l,ij} - \left(\mathbf{R}_{l,ij}^{(0)}\right)^2\right\|^2\right], \tag{35a}$$

$$\mathcal{J}_{\mathrm{KE}}(\boldsymbol{\theta}) = \mathbb{E}_{t,\boldsymbol{x}_0,\boldsymbol{\epsilon}}\left[w_2(t)\sum_{l,k,d}\left\|\left(\tilde{\mathbf{V}}_{\boldsymbol{\theta}}\right)_{l,k,d} - \left(\mathbf{V}_{l,k,d}^{(0)}\right)^2\right\|^2\right]. \tag{35b}$$

The models $\tilde{\mathbf{R}}_{\boldsymbol{\theta}}$ and $\tilde{\mathbf{V}}_{\boldsymbol{\theta}}$ utilize the same hidden state as the model $\mathbf{X}_{\boldsymbol{\theta}}$, with $\mathbf{X}_{\boldsymbol{\theta}}$ responsible for predicting $\mathbb{E}[\mathbf{X}^{(0)} \mid \mathbf{X}^{(t)}]$. The underlying structure of $\mathbf{X}_{\boldsymbol{\theta}}$ is based on EGNN for extracting node and edge features and a GRU network for dealing with time series. As a result, the outputs of $\tilde{\mathbf{R}}_{\boldsymbol{\theta}}$ are produced by an additional linear layer that processes the edge features generated by EGNN within $\mathbf{X}_{\boldsymbol{\theta}}$, and the outputs of $\tilde{\mathbf{V}}_{\boldsymbol{\theta}}$ are produced by an additional linear layer that takes the hidden state of the GRU within $\mathbf{X}_{\boldsymbol{\theta}}$ as input.

### E.2 DETAILS OF EXPERIMENT RESULTS

Tab. 8 and Tab. 9 present the outcomes of the grid search conducted on both the three-body and five-spring datasets. For the three-body datasets, the top three combinations of hyperparameters—hidden size and the number of layers—are highlighted for each training method. For the five-spring datasets, the top three hidden size hyperparameters identified for each training method are provided.

Table 8: The outcomes of the grid search conducted on the three-body datasets are summarized. For each training method, we highlight the top three combinations of hyperparameters, focusing on hidden size and the number of layers. "linear" refers to the settings in 10, and "reducible nonlinear" and "general nonlinear" refer to the settings in Equation 33 Equation 34, respectively. We first perform a grid search to find the optimal hyperparameters. Then, we conduct experiments with the optimal parameters, running each experiment five times using the seeds 42, 0, 1, 2, and 3. However, we observe that using seed 2 for noise matching results in poor performance, so we exclude this trial from all methods.

| Method | Hyperparameter | Trajectory error | Velocity error | Energy error |
|---|---|---|---|---|
| data matching | 256, 4 | 5.2455 | 4.2028 | 12.758 |
| | 512, 5 | 5.7765 | 3.8985 | 13.636 |
| | 256, 5 | 5.5098 | 4.4144 | 11.643 |
| noise matching | 256, 4 | 2.4132±0.1208 | 2.5745±0.0790 | 4.3292±0.7235 |
| | 256, 5 | 2.5695 | 2.6713 | 3.8944 |
| | 512, 3 | 2.6368 | 2.7192 | 3.5427 |
| noise matching + conservation of momentum (linear) | 512, 5 | 2.1261±0.0533 | 2.2815±0.0340 | 3.8698±0.2486 |
| | 1024, 4 | 2.4179 | 2.5261 | 3.9003 |
| | 512, 4 | 2.4188 | 2.5264 | 6.8971 |
| noise matching + conservation of energy (reducible nonlinear) | 128, 3 | **1.9880±0.3418** | **0.8328±0.1042** | **0.5465±0.0705** |
| | 128, 4 | **1.6659** | **0.7605** | **0.5198** |
| | 128, 5 | **1.7821** | **0.8030** | **0.4532** |
| noise matching + conservation of energy (general nonlinear) | 512, 4 | 2.2166±0.0669 | 2.4089±0.0941 | 4.5945±0.3960 |
| | 512, 3 | 2.5335 | 2.6234 | 3.8091 |
| | 1024, 3 | 2.5068 | 2.6737 | 5.2131 |

### E.3 SENSITIVITY OF PENALTY LOSS WEIGHT

Regarding the hyperparameters of the penalty loss weight, our experimental results show that the model performance is not highly sensitive to variations in the loss weight. Specifically, we conducted a search for the loss weight across a logarithmic scale, testing it approximately five times. To further illustrate this, we provide an example of how the loss weight affects model performance on the three-body dataset using conservation of momentum in Fig. 8.

Table 9: Grid search results for the five-spring datasets are provided. We present the top two hyper-parameter hidden size identified for each training method. The results of our experiment indicate that the output of the GRU model is inadequate to accurately predict the distance between particles. This limitation renders the methods designed for reducible nonlinear cases inapplicable, and consequently, their results have been excluded from our study. In contrast, the convoluted edge features generated by the EGNN model are sufficiently informative for predicting particle distances. Moreover, the application of methods suitable for general nonlinear cases improves performance. "linear" refers to the settings in Equation 10 and "general nonlinear" refers to the settings in Equation 35.

| Method | Hyperparameter | Dynamic error ($\times 10^{-2}$) | Momentum error | Energy error |
|---|---|---|---|---|
| data matching | 1024 | 5.3120 | 5.2320 | 1.1204 |
| | 256 | 5.3872 | 5.1448 | 1.1030 |
| noise matching | 512 | 5.1754±0.0286 | 5.3699±0.0462 | 1.0618±0.0243 |
| | 256 | 5.1950 | 5.3468 | 1.0805 |
| noise matching + conservation of momentum (linear) | 256 | **5.0731±0.0406** | **0.3898±0.0118** | **0.7418±0.0129** |
| | 512 | **5.0990** | **0.4335** | **0.7652** |
| noise matching + conservation of energy (general nonlinear) | 256 | 5.1643±0.0528 | 5.3359±0.0811 | 1.0500±0.0473 |
| | 1024 | 5.1809 | 5.3902 | 1.0879 |

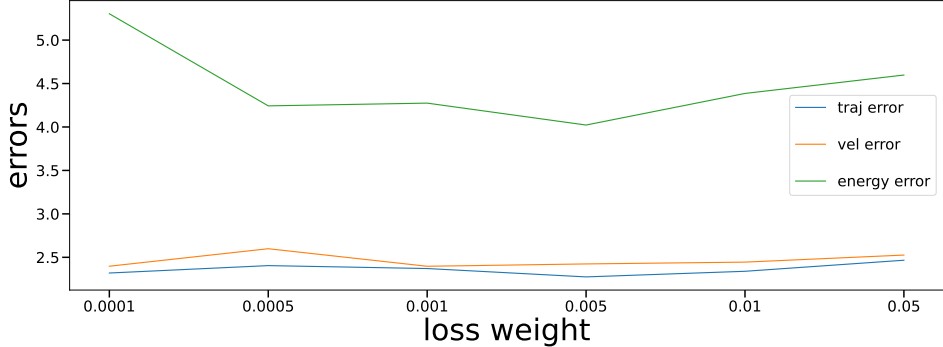

Figure 8: Sensitivity of model performance to variations in the penalty loss weight on the three-body dataset. The table shows the results of testing different loss weights on a logarithmic scale, with performance metrics recorded at five distinct points. The results indicate that model performance is relatively stable across a range of loss weight values.

### E.4 COMPARSION WITH PREDICTION METHODS

We conduct a comparison between the performance of the generation and prediction methods. In Tab. 10, we present a comparative analysis of generative models and prediction models for predicting physical dynamics, specifically advection and Darcy flow. The results of the prediction methods are taken from Takamoto et al. (2022), which performs a comprehensive comparison of FNO (Li et al., 2020), Unet (Ronneberger et al., 2015), and PINN (Raissi et al., 2019) (using DeepXDE (Lu et al., 2021)). Generative models that use diffusion techniques, both with and without prior information, exhibit comparable performance in both tasks. The diffusion model with priors shows an improvement over the one without priors. In this work, we do not conduct the procedure of super-resolution or denoising (Wang et al., 2018; Saharia et al., 2022), which are critical in practical applications to produce high-quality, clean, and detailed images from diffusion models. Hence, the performance of diffusion models can be further enhanced by the introduction of super-resolution and denoising.

Despite the inherent minor noise in the data, the model successfully captures the underlying patterns of the system, identifying low-frequency components and representing broad trends over time. However, challenges arise in accurately modeling high-frequency details, especially in highly non-linear systems (Darcy flow dataset). This limitation becomes apparent in multi-step forecasting, where errors in resolving finer details can accumulate, leading to significant deviations from the true system behavior as the prediction horizon extends (Lippe et al., 2024).

Table 10: Performance comparison of diffusion generative models with prediction models. The results of the prediction methods are brought from Takamoto et al. (2022).

| | Method | Backbone | Advection | Darcy flow |
|---|---|---|---|---|
| Generation | diffusion w/o prior | Karras Unet | $1.716 \times 10^{-2}$ | $2.016 \times 10^{-2}$ |
| | diffusion w/ prior | | $1.621 \times 10^{-2}$ | $1.954 \times 10^{-2}$ |
| Prediction | forward propagator approximation | FNO | $\mathbf{4.9 \times 10^{-3}}$ | $1.2 \times 10^{-2}$ |
| | autoregressive method | Unet | $3.8 \times 10^{-2}$ | $\mathbf{6.4 \times 10^{-3}}$ |
| | PINN | DeepXDE | $7.8 \times 10^{-1}$ | - |

We also test how predictive models perform on the five-spring dataset. We test the performance of NRI decoder in Kipf et al. (2018), which is specially designed for the dynamics and interacting systems. The results can be seen in Tab. 11.

Table 11: Performance comparison of diffusion generative models with prediction models on the five-spring dataset. We use the model (decoder only) in Kipf et al. (2018) to predict the dynamics and use EGNN (Satorras et al., 2021) and GRU (Chung et al., 2014) for diffusion methods. The metric of dynamic error is different from those in other tables. For other tables, we use the mean of the error between the generated dynamics and the ground truth dynamics solved by finite element methods in the following 8 steps, while in this table, we use only 1 step. The definition of the momentum error and energy error are the same. Note that diffusion methods cannot surpass predictive methods without priors. After the incorporation of the priors, diffusion models generate high-quality samples compared with the predictive methods.

| Method | Dynamic error ($\times 10^{-3}$) | Momentum error | Energy error |
|---|---|---|---|
| noise matching | 2.5178 | 5.3511 | 1.0891 |
| noise matching + priors | **2.4329** | **0.3687** | **0.7448** |
| NRI predict | 2.4471 | 5.2234 | 1.4577 |

### E.5  WHY NOT TRANSFORMER?

We attempted to implement a transformer architecture as the backbone for sequential data in particle dynamics datasets. However, our results indicate that the transformer-based model does not achieve performance comparable to that of recurrent structure backbones. This discrepancy is likely due to the nature of physical dynamics, where the next state is strongly dependent on the current state. The attention mechanism employed by transformers may reduce performance in this context, as it does not inherently account for the temporal evolution of states.

### E.6  ABLATION STUDIES ON DISTRIBUTIONAL PRIOR THROUGH DATA AUGMENTATION

Some existing works argue that considering the invariance property of the distribution may sacrifice the empirical performance(Yan et al., 2023). We perform further ablation studies to examine the significance of the distributional priors. Specifically, we evaluate two scenarios: 1) with permutational data augmentation, and 2) without permutational augmentation. To perform the permutational data augmentation, we randomly permute particles along with other properties such as their edge connectivity. The results are presented in Tab. 12 and 13 below.

Table 12: Ablation study on data augmentation for the equivalence of the model on the three-body dataset.

| Method | Traj error | | Velocity error | | Energy error | |
|---|---|---|---|---|---|---|
| | w/o aug | w/ aug | w/o aug | w/ aug | w/o aug | w/ aug |
| momentum conservation | 2.1953 | **2.1409** | 2.2974 | **2.2529** | **3.4364** | 4.1116 |
| reducible energy conservation | 1.8170 | **1.6072** | 1.9239 | **0.7307** | 3.2382 | **0.5062** |

We also test the equivalence of the trained models, one trained with data augmentation and the other without. Results show that, even without the introduction of data augmentation for equivalence, the

Table 13: Ablation study on data augmentation for the equivalence of the model on the five-spring dataset.

| Method | Dynamic error | | Momentum error | | Energy error | |
|---|---|---|---|---|---|---|
| | w/o aug | w/ aug | w/o aug | w/ aug | w/o aug | w/ aug |
| momentum conservation | 6.3972 | **5.0919** | **0.3564** | 0.3687 | 1.5536 | **0.7448** |
| reducible energy conservation | 6.6211 | **5.1615** | 7.0716 | **5.3032** | 2.4910 | **1.0548** |

model still learns to exhibit the desired equivalence, which aligns with our mathematical analysis. Furthermore, when equivalence data augmentation is applied, the model achieves a significant reduction in equivalence error, decreasing from 1.6e-3 to 3.5e-4. This further supports the correctness of our analysis, demonstrating that, when using two models with the same capacity, a $\left(\mathcal{G}, \nabla^{-1}\right)$-equivariant model should be employed for noise matching.

### E.7 SAMPLING IN FEWER STEPS USING DPM-SOLVERS

We conduct experiments of using the DPM-solvers (Lu et al., 2022) to sample in fewer steps. By reducing the number of diffusion steps required, DPM solvers significantly lower computational expenses in generating physics dynamics. This efficiency is achieved with minimal degradation in performance, ensuring that the resulting dynamics remain closely aligned with the underlying physical principles. We apply the DPM-Solver-3 (Algorithm 2 in Lu et al. (2022)) and the results can be seen in Fig. 9 and 10.

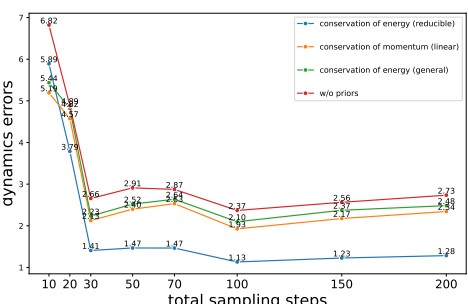
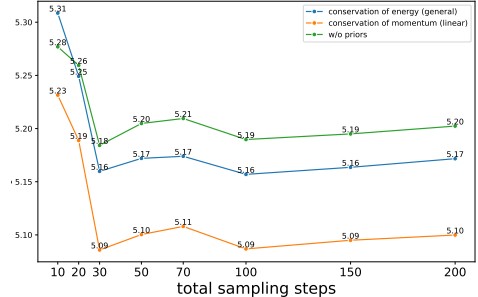

Figure 9: Results of sampling through DPM-Solver-3 on the three-body dataset. The x-axis denotes the values for $M$ in Algorithm 2 in Lu et al. (2022).

Figure 10: Results of sampling through DPM-Solver-3 on the five-spring dataset. The x-axis denotes the values for $M$ in Algorithm 2 in Lu et al. (2022).

## F PROOFS

### F.1 SUFFICIENT CONDITIONS FOR THE INVARIANCE OF MARGINAL DISTRIBUTION

**Definition 5** (volume-preserving). *A function whose derivative has a determinant equal to 1 is known as a **volume-preserving** function.*

**Definition 6** (isomorphism). *An **isomorphism** is a structure-preserving mapping between two structures of the same type that can be reversed by an inverse mapping.*

**Definition 7** (diffeomorphism). *A **diffeomorphism** is an isomorphism of differentiable manifolds. It is an invertible function that maps one differentiable manifold to another such that both the function and its inverse are continuously differentiable.*

**Definition 8** (isometry). *Let $\mathbf{X}$ and $\mathbf{Y}$ be metric spaces with metrics (e.g., distances) $d_{\mathbf{X}}$ and $d_{\mathbf{Y}}$. A map $\mathbf{f} : \mathbf{X} \rightarrow \mathbf{Y}$ is called an **isometry** if for any $\mathbf{a}, \mathbf{b} \in \mathbf{X}$, $d_{\mathbf{X}}(\mathbf{a}, \mathbf{b}) = d_{\mathbf{Y}}(\mathbf{f}(\mathbf{a}), \mathbf{f}(\mathbf{b}))$.*

**Definition 9** (homothety). *If for all $\mathbf{G} \in \mathcal{G}$ and for all scalar $0 < a < 1$, there exists $\mathbf{H} \in \mathcal{G}$ such that $\mathbf{H}(a\boldsymbol{x}) = a\mathbf{G}(\boldsymbol{x})$. $\mathbf{H}$ would be a **homothety** (a transformation that scales distances*

*by a constant factor but does not necessarily preserve angles). The group $\mathcal{G}$ formed by all such transformations is called the **homothety group**.*

**Theorem 1** (Sufficient conditions for the invariance of $q_0$ to imply the invariance of $q_t$). *Let $q_0$ be a $\mathcal{G}$-invariant distribution. If for all $\mathbf{G} \in \mathcal{G}$, $\mathbf{G}$ is volume-preserving diffeomorphism and isometry, and for all $0 < a < 1$, there exists $\mathbf{H} \in \mathcal{G}$ such that $\mathbf{H}(a\boldsymbol{x}) = a\mathbf{G}(\boldsymbol{x})$, then $q_t$ is also $\mathcal{G}$-invariant.*

*Proof.* For VE diffusion (defined in Sec. 3.4 in Song et al. (2020)), $q_t(\boldsymbol{x}_t \mid \boldsymbol{x}_0) = \mathcal{N}(\boldsymbol{x}_t \mid \boldsymbol{x}_0, \sigma_t^2 \mathbf{I})$. For any $\mathcal{G}$-invariant distribution $q_0$ and $\mathbf{G} \in \mathcal{G}$, we have

$$q_t(\mathbf{G}(\boldsymbol{x}_t)) = \int q_t(\mathbf{G}(\boldsymbol{x}_t) \mid \boldsymbol{x}_0) q_0(\boldsymbol{x}_0) \mathrm{d}\boldsymbol{x}_0 \qquad \text{probability chain rule} \tag{36a}$$

$$= \int q_t(\mathbf{G}(\boldsymbol{x}_t) \mid \mathbf{G}(\boldsymbol{x}_0)) q_0(\mathbf{G}(\boldsymbol{x}_0)) \mathrm{d}\mathbf{G}(\boldsymbol{x}_0) \qquad \text{change of variables} \tag{36b}$$

$$= \int \mathcal{N}(\mathbf{G}(\boldsymbol{x}_t) \mid \mathbf{G}(\boldsymbol{x}_0), \sigma_t^2 \mathbf{I}) q_0(\boldsymbol{x}_0) \mathrm{d}\boldsymbol{x}_0 \qquad \text{volume-preserving diffeomorphism} \tag{36c}$$

$$= \int \mathcal{N}(\boldsymbol{x}_t \mid \boldsymbol{x}_0, \sigma_t^2 \mathbf{I}) q_0(\boldsymbol{x}_0) \mathrm{d}\boldsymbol{x}_0 \qquad \text{isotropic Gaussian } \mathcal{N} \text{ and isometry } \mathbf{G} \tag{36d}$$

$$= \int q_t(\boldsymbol{x}_t \mid \boldsymbol{x}_0) q_0(\boldsymbol{x}_t) \mathrm{d}\boldsymbol{x}_0 \tag{36e}$$

$$= q_t(\boldsymbol{x}_t) \tag{36f}$$

Hence, the marginal distribution $q_t$ at any time $t$ is an $\mathcal{G}$-invariant distribution.

For VP diffusion (defined in Sec. 3.4 in Song et al. (2020)), assume $\alpha_t > 0$ at any time $t$. $q_t(\boldsymbol{x}_t \mid \boldsymbol{x}_0) = \mathcal{N}(\boldsymbol{x}_t \mid \alpha_t \boldsymbol{x}_0, (1 - \alpha_t)\mathbf{I})$. Define $\hat{q}_t(\boldsymbol{x}_t) = q_t(\frac{1}{\alpha_t}\boldsymbol{x}_t)$. Note that $\frac{1}{\alpha_t}\boldsymbol{x}_t = \boldsymbol{x}_0 + \frac{\sqrt{1-\alpha_t}}{\alpha_t}\boldsymbol{\epsilon}, \boldsymbol{\epsilon} \sim \mathcal{N}(\mathbf{0}, \mathbf{I})$, is a random variable generated by some VE diffusion process. Hence, its marginal distribution at any time $t$ is $\mathcal{G}$-invariant. For any $\mathcal{G}$-invariant distribution $q_0$, we have

$$q_t(\mathbf{G}(\boldsymbol{x}_t)) = \hat{q}_t(\alpha_t \mathbf{G}(\boldsymbol{x}_t)) \qquad \text{by definition} \tag{37a}$$

$$= \hat{q}_t(\mathbf{H}(\alpha_t \boldsymbol{x}_t)) \qquad \text{by sufficient conditions} \tag{37b}$$

$$= \hat{q}_t(\alpha_t \boldsymbol{x}_t) \qquad \mathcal{G}\text{-invariance} \tag{37c}$$

$$= q_t(\boldsymbol{x}_t) \tag{37d}$$

$\square$

**Discussion of related theorems.**

**Theorem 10** (Proposition 1 in Xu et al. (2022)). *Let $q(\boldsymbol{x}_T)$ be an SE(3)-invariant density function, i.e., $q(\boldsymbol{x}_T) = q(\mathbf{G}(\boldsymbol{x}_T))$ for $\mathbf{G} \in \mathrm{SE}(3)$. If Markov transitions $q(\boldsymbol{x}_{t-1} \mid \boldsymbol{x}_t)$ are SE(3)-equivariant, i.e., $q(\boldsymbol{x}_{t-1} \mid \boldsymbol{x}_t) = q(\mathbf{G}(\boldsymbol{x}_{t-1}) \mid \mathbf{G}(\boldsymbol{x}_t))$, then we have that the density $q_\theta(\boldsymbol{x}_0) = \int q(\boldsymbol{x}_T) q_\theta(\boldsymbol{x}_{0:T-1} \mid \boldsymbol{x}_T) \mathrm{d}\boldsymbol{x}_{1:T}$ is also SE(3)-invariant.*

Xu et al. (2022) explore the integration of invariance during the sampling process while disregarding it during the forward process. Xu et al. (2022) propose that sampling through equivalent translational kernel results in invariant distributions. In contrast, our Theorem 1 demonstrates that even when the transition probabilities of the Markov chain are not SE(n)-equivariant, the resulting composed distribution can still be SE(n)-invariant. This result offers a stronger conclusion than that presented by Xu et al. (2022).

**Theorem 11** (Proposition 3.6 in Yim et al. (2023), Proposition 3.1 in Mathieu et al. (2024)). *Let $\mathcal{G}$ be a Lie group and $\mathcal{H}$ a subgroup of $\mathcal{G}$. Let $\mathbf{X}^{(0)} \sim q_0$ for an $\mathcal{H}$ invariant distribution $q_0$. If $\mathrm{d}\mathbf{X}^{(t)} = b(t, \mathbf{X}^{(t)}) \mathrm{d}t + \Sigma^{1/2} \mathrm{d}\mathbf{B}^{(t)}$ for bounded, $\mathcal{H}$-equivariant coefficients $b$ and $\Sigma$ satisfying $b \circ L_h = \mathrm{d}L_h(b)$ and $\Sigma \mathrm{d}L_h(\cdot) = \mathrm{d}L_h(\Sigma \cdot)$, and where $\mathbf{B}^{(t)}$ is a Brownian motion associated with a left-invariant metric. Then the distribution $q_t$ of $\mathbf{X}^{(t)}$ is $\mathcal{H}$-invariant.*

In contrast to our Theorem 1, Theorem 11 in Yim et al. (2023) and Mathieu et al. (2024) imposes specific conditions on the relationship between the forward diffusion scheduler and the group operators, whereas our theorem does not. However, Yim et al. (2023) impose fewer constraints on the properties of the group operations.

### F.2 INVARIANT DISTRIBUTION EXAMPLES

#### F.2.1 SE(n)-INVARIANT DISTRIBUTION

If $q_0$ is an SE(n)-invariant distribution, then $q_t$ is also SE(n)-invariant.

*Proof.* Given any $\mathbf{G} \in \mathrm{SE(n)}$, let $\mathbf{G}(\boldsymbol{x}) = \mathbf{R}\boldsymbol{x} + \mathbf{b}$, where $\mathbf{R} \in \mathrm{SO}(n), \mathbf{b} \in \mathbb{R}^n$.

- **volume-preserving**: $\det\left(\frac{\mathrm{d}\mathbf{G}(\boldsymbol{x})}{\mathrm{d}\boldsymbol{x}}\right) = \det\left(\mathbf{R}^\top\right) = 1$.

- **diffeomorphism**: *smoothness*: The transformation $\mathbf{G}(\boldsymbol{x}) = \mathbf{R}\boldsymbol{x} + \mathbf{b}$ is smooth because it involves linear operations (rotation and translation) that are smooth. Specifically, the rotation $\mathbf{R}$ and the translation $\mathbf{b}$ are smooth functions of their parameters; *bijectivity*: The function $\mathbf{G}(\boldsymbol{x})$ is bijective. For any $\boldsymbol{x} \in \mathbb{R}^n$, the function $\mathbf{G}(\boldsymbol{x})$ is one-to-one and onto. The inverse function is given by: $\mathbf{G}^{-1}(\boldsymbol{y}) = \mathbf{R}^{-1}(\boldsymbol{y} - \mathbf{b})$, where $\boldsymbol{y} \in \mathbb{R}^n$. Since $\mathbf{R}$ is a rotation matrix, it is invertible, and its inverse $\mathbf{R}^{-1}$ is also smooth. Therefore, the inverse function is smooth.

- **isometry**: for all $\boldsymbol{x}, \boldsymbol{y} \in \mathbb{R}^n, \|\mathbf{G}(\boldsymbol{x}) - \mathbf{G}(\boldsymbol{y})\|^2 = \|\mathbf{R}\boldsymbol{x} + \mathbf{b} - (\mathbf{R}\boldsymbol{y} + \mathbf{b})\|^2 = (\boldsymbol{x} - \boldsymbol{y})^\top \mathbf{R}^\top \mathbf{R}(\boldsymbol{x} - \boldsymbol{y}) = \|\boldsymbol{x} - \boldsymbol{y}\|^2$. Hence, $\|\mathbf{G}(\boldsymbol{x}) - \mathbf{G}(\boldsymbol{y})\| = \|\boldsymbol{x} - \boldsymbol{y}\|$.

- **homothety**: Given any $\mathbf{G} \in \mathrm{SE(n)}$ and $0 < a < 1$, let $\mathbf{H}(\boldsymbol{x}) = \mathbf{R}\boldsymbol{x} + a\mathbf{b} \in \mathrm{SE(n)}$. Then, $\mathbf{H}(a\boldsymbol{x}) = \mathbf{R}(a\boldsymbol{x}) + a\mathbf{b} = a(\mathbf{R}\boldsymbol{x} + \mathbf{b}) = a\mathbf{G}(\boldsymbol{x})$.

Hence, sufficient conditions are satisfied and $q_t$ is also SE(n)-invariant. $\square$

Let $q$ be an SE(n)-invariant distribution. Given a set of $m$ points $\mathcal{C} \in \mathbb{R}^{n \times m}$, we write it in the vector form $\boldsymbol{x} := \mathrm{vec}(\mathcal{C}) \in \mathbb{R}^{mn}$. For any $\mathbf{G}(\mathcal{C}) = \mathbf{R}\mathcal{C} + \mathbf{b}, \mathbf{R} \in \mathrm{SO}(n)$, let $\hat{\mathbf{R}} \in \mathbb{R}^{mn \times mn}$ be a block matrix with $\mathbf{R}$ on its diagonal block and $\hat{\mathbf{b}} = [\mathbf{b}^\top \cdots, \mathbf{b}^\top]^\top \in \mathbb{R}^{mn}$. We have $\hat{\mathbf{G}}(\boldsymbol{x}) = \hat{\mathbf{R}}\boldsymbol{x} + \hat{\mathbf{b}}$. Then, $(\nabla_{\boldsymbol{x}}\hat{\mathbf{G}}(\boldsymbol{x}))^{-1} = (\hat{\mathbf{R}}^\top)^{-1} = \hat{\mathbf{R}}$. Hence, $\nabla_{\hat{\mathbf{G}}(\boldsymbol{x})} \log q(\hat{\mathbf{G}}(\boldsymbol{x})) = \hat{\mathbf{R}}\nabla_{\boldsymbol{x}} \log q(\boldsymbol{x})$, which imples $\nabla_{\mathbf{G}(\mathcal{C})} \log q(\mathbf{G}(\mathcal{C})) = \mathbf{R}\nabla_{\mathcal{C}} \log q(\mathcal{C})$. Thus, the score function of an SE(n)-invariant distribution is SO(n)-equivariant and translational-invariant.

#### F.2.2 PERMUTATION-INVARIANT DISTRIBUTION

We first list some useful properties of the Kronecker product:

- $\mathrm{vec}(\mathbf{A}\mathbf{X}\mathbf{B}) = (\mathbf{B}^T \otimes \mathbf{A})\mathrm{vec}(\mathbf{X})$.
- $\det(\mathbf{A} \otimes \mathbf{B}) = \det(\mathbf{A})^m \det(\mathbf{B})^n$, where $\mathbf{A} \in \mathbb{R}^{m \times m}, \mathbf{B} \in \mathbb{R}^{n \times n}$.
- $(\mathbf{A} \otimes \mathbf{B})^T = \mathbf{A}^T \otimes \mathbf{B}^T$.
- $(\mathbf{A} \otimes \mathbf{B})(\mathbf{C} \otimes \mathbf{D}) = (\mathbf{A}\mathbf{C}) \otimes (\mathbf{B}\mathbf{D})$.
- For square nonsingular matrices $\mathbf{A}$ and $\mathbf{B}$: $(\mathbf{A} \otimes \mathbf{B})^{-1} = \mathbf{A}^{-1} \otimes \mathbf{B}^{-1}$.

If $q_0$ is a permutation-invariant, then $q_t$ is also permutation-invariant.

*Proof.* Given any $\mathbf{G} \in \mathcal{G}$ with $\mathbf{G}(\mathbf{X}) = \mathbf{P}\mathbf{X}\mathbf{P}^\top$, we consider its vector form of $\mathrm{vec}(\mathbf{G}(\mathbf{X})) = \mathrm{vec}(\mathbf{P}\mathbf{X}\mathbf{P}^\top) = (\mathbf{P} \otimes \mathbf{P})\mathrm{vec}(\mathbf{X})$.

- **volume-preserving**: $\det\left(\frac{\mathrm{d}\,\mathrm{vec}(\mathbf{G}(\mathbf{X}))}{\mathrm{d}\,\mathrm{vec}(\mathbf{X})}\right) = \det\left((\mathbf{P} \otimes \mathbf{P})^\top\right) = \det(\mathbf{P} \otimes \mathbf{P}) = \det(\mathbf{P})^n \det(\mathbf{P})^n = \det(\mathbf{P})^{2n} = (\pm 1)^{2n} = 1$.

- **diffeomorphism**: *smoothness*: $\mathbf{G}$ is smooth because it involves matrix multiplication, which is a smooth operation in $\mathbf{X}$. Since $\mathbf{P}$ and $\mathbf{P}^\top$ are constant matrices (not functions of $\mathbf{X}$), $\mathbf{G}$ inherits the smoothness from the matrix operations; *bijectivity*: Suppose $\mathbf{Y} = \mathbf{G}(\mathbf{X}) = \mathbf{P}\mathbf{X}\mathbf{P}^\top$. To recover $\mathbf{X}$ from $\mathbf{Y}$, we compute: $\mathbf{X} = \mathbf{P}^\top\mathbf{Y}\mathbf{P}$, which is a valid operation because $\mathbf{P}$ is invertible.

- **isometry**: note that $(\mathbf{P} \otimes \mathbf{P})^\top (\mathbf{P} \otimes \mathbf{P}) = (\mathbf{P}^\top \otimes \mathbf{P}^\top) (\mathbf{P} \otimes \mathbf{P}) = (\mathbf{P}^\top\mathbf{P}) \otimes (\mathbf{P}^\top\mathbf{P}) = \mathbf{I} \otimes \mathbf{I} = \mathbf{I}$. For all $\mathbf{X}, \mathbf{Y} \in \mathbb{R}^{n \times n}$,

$$\| \operatorname{vec}(\mathbf{G}(\mathbf{X})) - \operatorname{vec}(\mathbf{G}(\mathbf{Y})) \|^2 \tag{38a}$$

$$= \| (\mathbf{P} \otimes \mathbf{P}) \operatorname{vec}(\mathbf{X}) - (\mathbf{P} \otimes \mathbf{P}) \operatorname{vec}(\mathbf{Y}) \|^2 \tag{38b}$$

$$= (\operatorname{vec}(\mathbf{X}) - \operatorname{vec}(\mathbf{Y}))^\top (\mathbf{P} \otimes \mathbf{P})^\top (\mathbf{P} \otimes \mathbf{P}) (\operatorname{vec}(\mathbf{X}) - \operatorname{vec}(\mathbf{Y})) \tag{38c}$$

$$= (\operatorname{vec}(\mathbf{X}) - \operatorname{vec}(\mathbf{Y}))^\top \mathbf{I} (\operatorname{vec}(\mathbf{X}) - \operatorname{vec}(\mathbf{Y})) \tag{38d}$$

$$= \| \operatorname{vec}(\mathbf{X}) - \operatorname{vec}(\mathbf{Y}) \|^2 \tag{38e}$$

Hence, $\|\mathbf{G}(\mathbf{X}) - \mathbf{G}(\mathbf{Y})\|_{\mathrm{F}} = \|\mathbf{X} - \mathbf{Y}\|_{\mathrm{F}}$.

- **homothety**: Given any $\mathbf{G} \in \mathcal{G}$ and $0 < a < 1$, let $\mathbf{H} = \mathbf{G} \in \mathcal{G}$. Then, $\mathbf{H}(a\mathbf{X}) = \mathbf{P}(a\mathbf{X})\mathbf{P}^\top = a\mathbf{P}\mathbf{X}\mathbf{P}^\top = a\mathbf{G}(\mathbf{X})$.

Hence, sufficient conditions are satisfied and $q_t$ is also permutation-invariant. $\square$

Let $q$ be a permutation-invariant distribution of feature $\mathbf{X} \in \mathbb{R}^{n \times n}$ such as affinity/connectivity matrices representing relationships or connections between pairs of entities (e.g., nodes in a graph) or Gram matrices in kernel methods representing similarities between a set of vectors. Let $q(\mathbf{P}\mathbf{X}\mathbf{P}^\top) = q(\mathbf{X})$ for any permutational matrix $\mathbf{P}$. Consider the vectorization of $\mathbf{X}$. Let $\hat{q}(\operatorname{vec}(\mathbf{X})) := q(\mathbf{X})$. Note that $\operatorname{vec}(\mathbf{P}\mathbf{X}\mathbf{P}^\top) = (\mathbf{P} \otimes \mathbf{P})\operatorname{vec}(\mathbf{X})$. Hence, $\nabla_{\operatorname{vec}(\mathbf{P}\mathbf{X}\mathbf{P}^\top)} \log \hat{q}(\operatorname{vec}(\mathbf{P}\mathbf{X}\mathbf{P}^\top)) = (\mathbf{P} \otimes \mathbf{P})^{-T}\nabla_{\operatorname{vec}(\mathbf{X})} \log \hat{q}(\operatorname{vec}(\mathbf{X})) = (\mathbf{P} \otimes \mathbf{P})\nabla_{\operatorname{vec}(\mathbf{X})} \log \hat{q}(\operatorname{vec}(\mathbf{X}))$. This implies that $\nabla_{\mathbf{P}\mathbf{X}\mathbf{P}^\top} \log q(\mathbf{P}\mathbf{X}\mathbf{P}^\top) = \mathbf{P}\nabla_{\mathbf{X}} \log q(\mathbf{X})\mathbf{P}^\top$. Thus, the score function of a permutation-invariant distribution is permutation-equivariant.

### F.3 ECM EQUIVALENCE

**Theorem 2** (Equivalence class manifold representation). *If we have a $(\mathcal{G}, \nabla^{-1})$-equivariant model such that $\mathbf{s}_{\boldsymbol{\theta}}(\boldsymbol{x}_t, t) = \nabla_{\boldsymbol{x}} \log q_{ECM}(\boldsymbol{x}_t)$ almost surely on $\boldsymbol{x}_t \in \mathrm{ECM}$, then we have $\mathbf{s}_{\boldsymbol{\theta}}(\boldsymbol{x}_t, t) = \nabla_{\boldsymbol{x}} \log q_t(\boldsymbol{x}_t)$ almost surely.*

*Proof.* Suppose we have a $(\mathcal{G}, \nabla^{-1})$-equivariant model $\mathbf{s}_{\boldsymbol{\theta}}$ and $\mathbf{s}_{\boldsymbol{\theta}}(\varphi(\boldsymbol{x})) = \nabla_{\varphi(\boldsymbol{x})} \log q_{ECM}(\varphi(\boldsymbol{x}))$ almost surely. Then, we have

$$\mathbf{s}_{\boldsymbol{\theta}}(\boldsymbol{x}) = \frac{\partial \varphi(\boldsymbol{x})}{\partial \boldsymbol{x}}\mathbf{s}_{\boldsymbol{\theta}}(\varphi(\boldsymbol{x})) \qquad \text{by equivariance of the model} \tag{39a}$$

$$= \frac{\partial \varphi(\boldsymbol{x})}{\partial \boldsymbol{x}}\nabla_{\varphi(\boldsymbol{x})} \log q_{ECM}(\varphi(\boldsymbol{x})) \tag{39b}$$

$$= \nabla_{\boldsymbol{x}} \log q_t(\boldsymbol{x}) \qquad \text{by Equation 5} \tag{39c}$$

$$\square$$

### F.4 MULTILINEAR JENSEN'S GAP

The following lemma is directly from the results of the optimal values of noise matching, which will be used in proving Theorem. 3.

**Lemma 12.** *The gradient of $\mathbb{E}_{t,\boldsymbol{x}_0,\boldsymbol{\epsilon}}[w(t)\|\boldsymbol{\epsilon}_{\boldsymbol{\theta}}(\boldsymbol{x}_t, t) - \boldsymbol{\epsilon}\|^2]$ w.r.t. $\boldsymbol{\theta}$ at $\boldsymbol{\epsilon}_{\boldsymbol{\theta}}(\boldsymbol{x}_t, t) = -\sigma_t\nabla_{\boldsymbol{x}} \log q_t(\boldsymbol{x}_t)$ equals $\mathbf{0}$.*

**Theorem 3** (Multilinear Jensen's gap). *The optimizer for $\mathbb{E}_{t,\boldsymbol{x}_0,\boldsymbol{\epsilon}}[w(t)\|\mathbf{u}_{\boldsymbol{\theta}_1}(\boldsymbol{x}_t, t) - \mathbf{u}_0\|^2]$ is the reweighted optimizer of $\mathbb{E}_{t,\boldsymbol{x}_0,\boldsymbol{\epsilon}}[w(t)\|\mathbf{W}_0\mathbf{u}_{\boldsymbol{\theta}_2}(\boldsymbol{x}_t, t) + \mathbf{b}_0\|^2]$ with reweighted variable $\mathbf{W}_0^\top\mathbf{W}_0$.*

*Proof.* Without loss of generality, suppose that the optimizer of $\mathbf{u}_{\boldsymbol{\theta}_2}$ is given by $\mathbf{u}_{\boldsymbol{\theta}_1^*} + \mathbf{u}_{\Delta\boldsymbol{\theta}}$. The loss optimizer of data matching is given by $\mathbf{u}_{\boldsymbol{\theta}_1^*}(\boldsymbol{x}_t, t) = \frac{1}{\alpha_t}\left(\boldsymbol{u}_t + \sigma_t^2 \nabla_{\boldsymbol{u}} \log q_t(\boldsymbol{x}_t)\right)$. Substituting into the PDE loss term, we have

$$\mathbb{E}_{t,\boldsymbol{x}_0,\boldsymbol{\epsilon}}[w(t)\|\mathbf{W}_0 \mathbf{u}_{\boldsymbol{\theta}_2}(\boldsymbol{x}_t, t) + \mathbf{b}_0\|^2] \tag{40a}$$

$$= \mathbb{E}_{t,\boldsymbol{x}_0,\boldsymbol{\epsilon}}[w(t)\|\frac{1}{\alpha_t}\mathbf{W}_0\left(\boldsymbol{u}_t + \sigma_t^2\nabla_{\boldsymbol{u}}\log q_t(\boldsymbol{x}_t)\right) + \mathbf{W}_0\mathbf{u}_{\Delta\boldsymbol{\theta}} + \mathbf{b}_0\|^2] \tag{40b}$$

$$= \mathbb{E}_{t,\boldsymbol{x}_0,\boldsymbol{\epsilon}}[w(t)\|\frac{1}{\alpha_t}\mathbf{W}_0\left(\alpha_t\boldsymbol{u}_0 + \sigma_t\boldsymbol{\epsilon} + \sigma_t^2\nabla_{\boldsymbol{u}}\log q_t(\boldsymbol{x}_t)\right) + \mathbf{W}_0\mathbf{u}_{\Delta\boldsymbol{\theta}} + \mathbf{b}_0\|^2] \tag{40c}$$

$$= \mathbb{E}_{t,\boldsymbol{x}_0,\boldsymbol{\epsilon}}[w(t)\|\mathbf{W}_0\boldsymbol{u}_0 + \mathbf{b}_0 + \frac{\sigma_t}{\alpha_t}\mathbf{W}_0\boldsymbol{\epsilon} + \frac{\sigma_t^2}{\alpha_t}\mathbf{W}_0\nabla_{\boldsymbol{u}}\log q_t(\boldsymbol{x}_t) + \mathbf{W}_0\mathbf{u}_{\Delta\boldsymbol{\theta}}\|^2] \tag{40d}$$

$$= \mathbb{E}_{t,\boldsymbol{x}_0,\boldsymbol{\epsilon}}[w(t)\|\frac{\sigma_t}{\alpha_t}\mathbf{W}_0\boldsymbol{\epsilon} + \frac{\sigma_t^2}{\alpha_t}\mathbf{W}_0\nabla_{\boldsymbol{u}}\log q_t(\boldsymbol{x}_t) + \mathbf{W}_0\mathbf{u}_{\Delta\boldsymbol{\theta}}\|^2] \tag{40e}$$

$$= \mathbb{E}_{t,\boldsymbol{x}_0,\boldsymbol{\epsilon}}[w(t)\frac{\sigma_t^2}{\alpha_t^2}\|\mathbf{W}_0\left(\boldsymbol{\epsilon} + \sigma_t\nabla_{\boldsymbol{u}}\log q_t(\boldsymbol{x}_t) + \frac{\alpha_t}{\sigma_t}\mathbf{u}_{\Delta\boldsymbol{\theta}}\right)\|^2] \tag{40f}$$

Dropping the reweighting term $\sigma_t^2/\alpha_t^2$ does not change the optimal solution. When $\mathbf{u}_{\Delta\boldsymbol{\theta}} \equiv \mathbf{0}$, observing the above objective and the noise matching objective, the above objective is the reweighted objective of noise matching by replacing $\mathbf{I}$ with $\mathbf{W}_0^\top\mathbf{W}_0$. $\square$

## G    VISUALIZATION OF GENERATED SAMPLES

In this study, we refrain from applying super-resolution and denoising procedures, which are essential in practical applications for generating high-quality, clear, and detailed images from diffusion models. Consequently, the generated samples contain noise. For the three-body dataset, since we only generate 10 frames, we apply the cubic spline to visualize a smooth trajectory and this is not applied when evaluating the quality of samples.

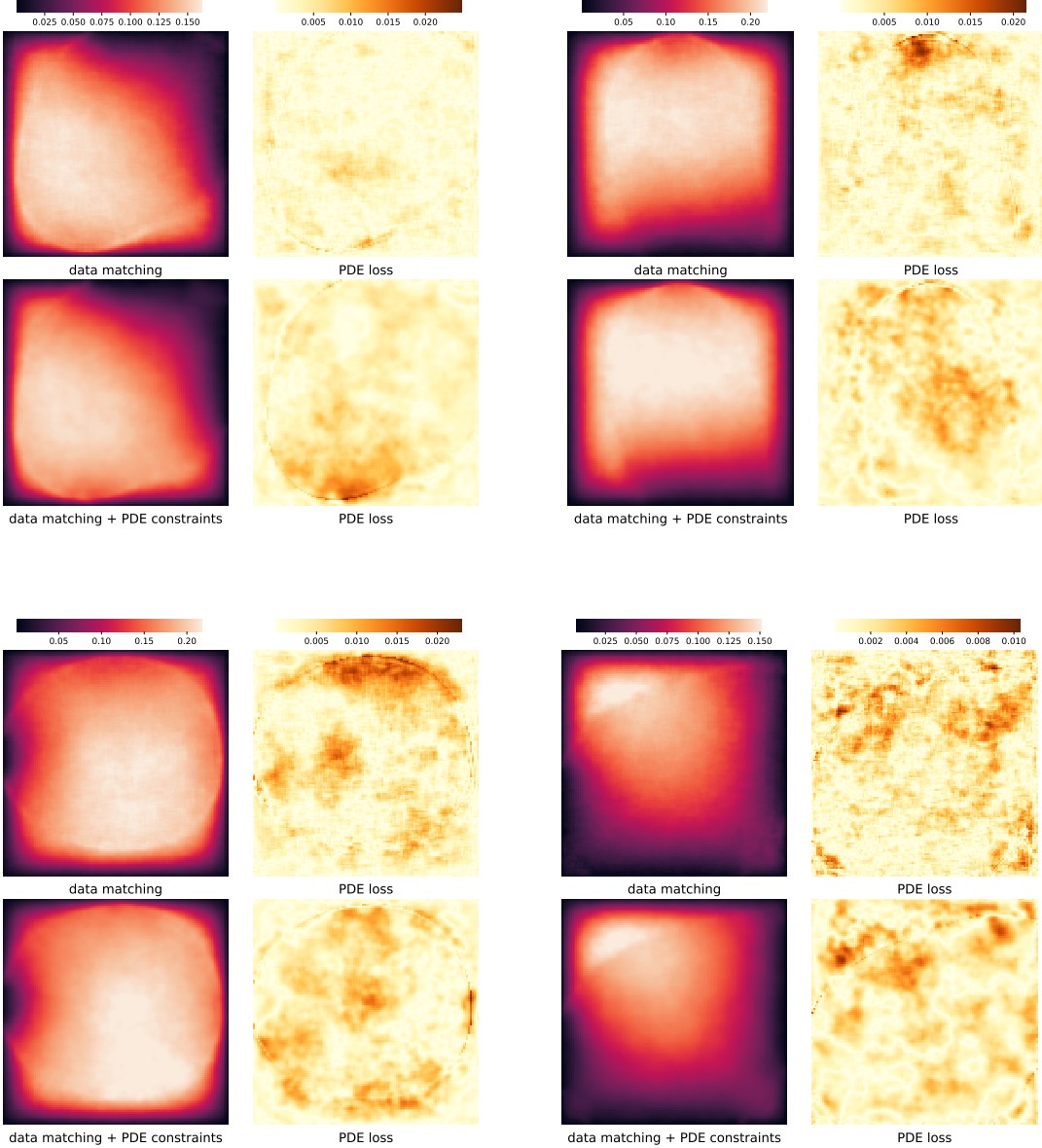

Figure 11: Examples of Darcy flow samples generated by models trained with/without PDE constraints.

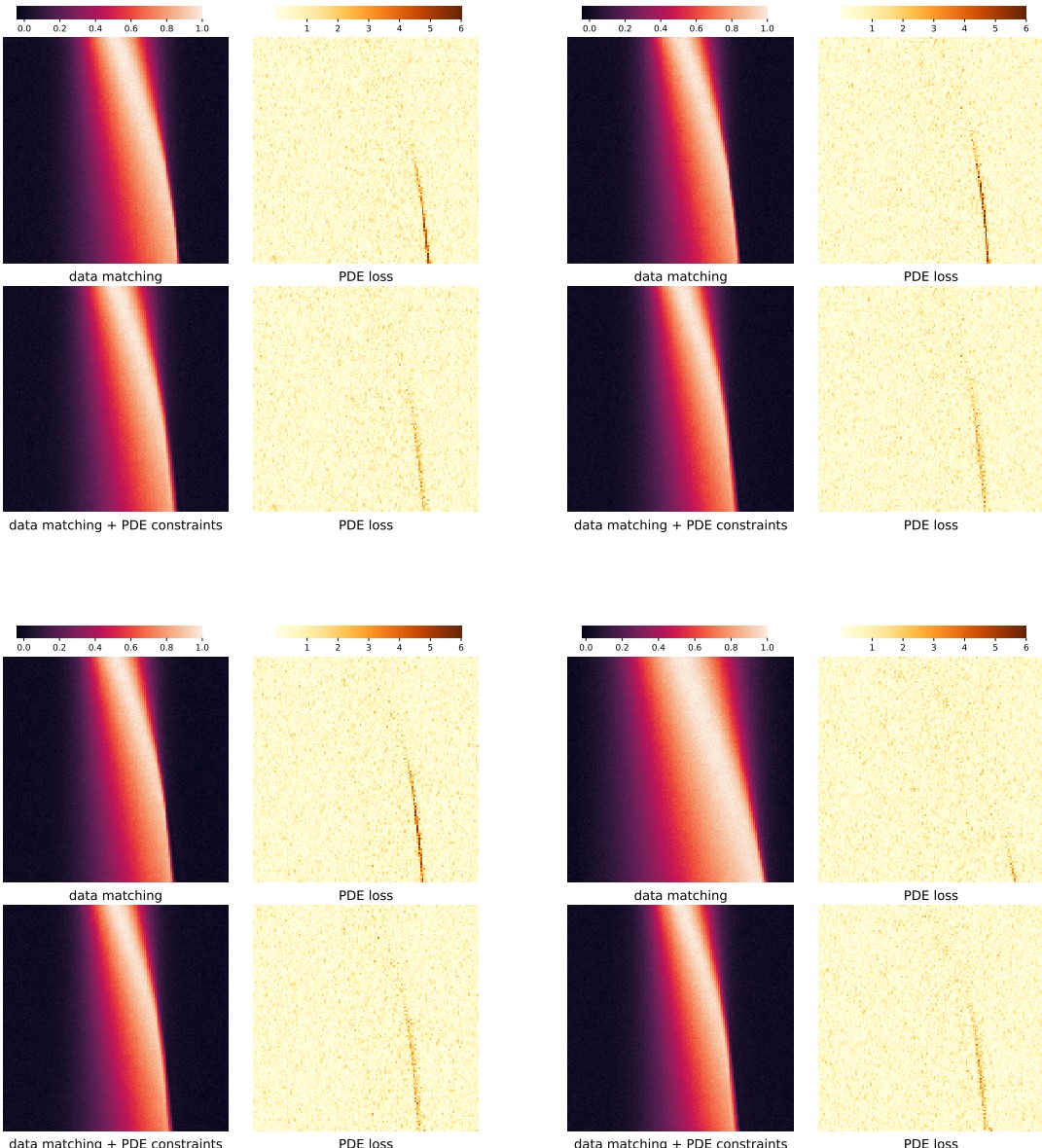

Figure 12: Examples of Burger samples generated by models trained with/without PDE constraints.

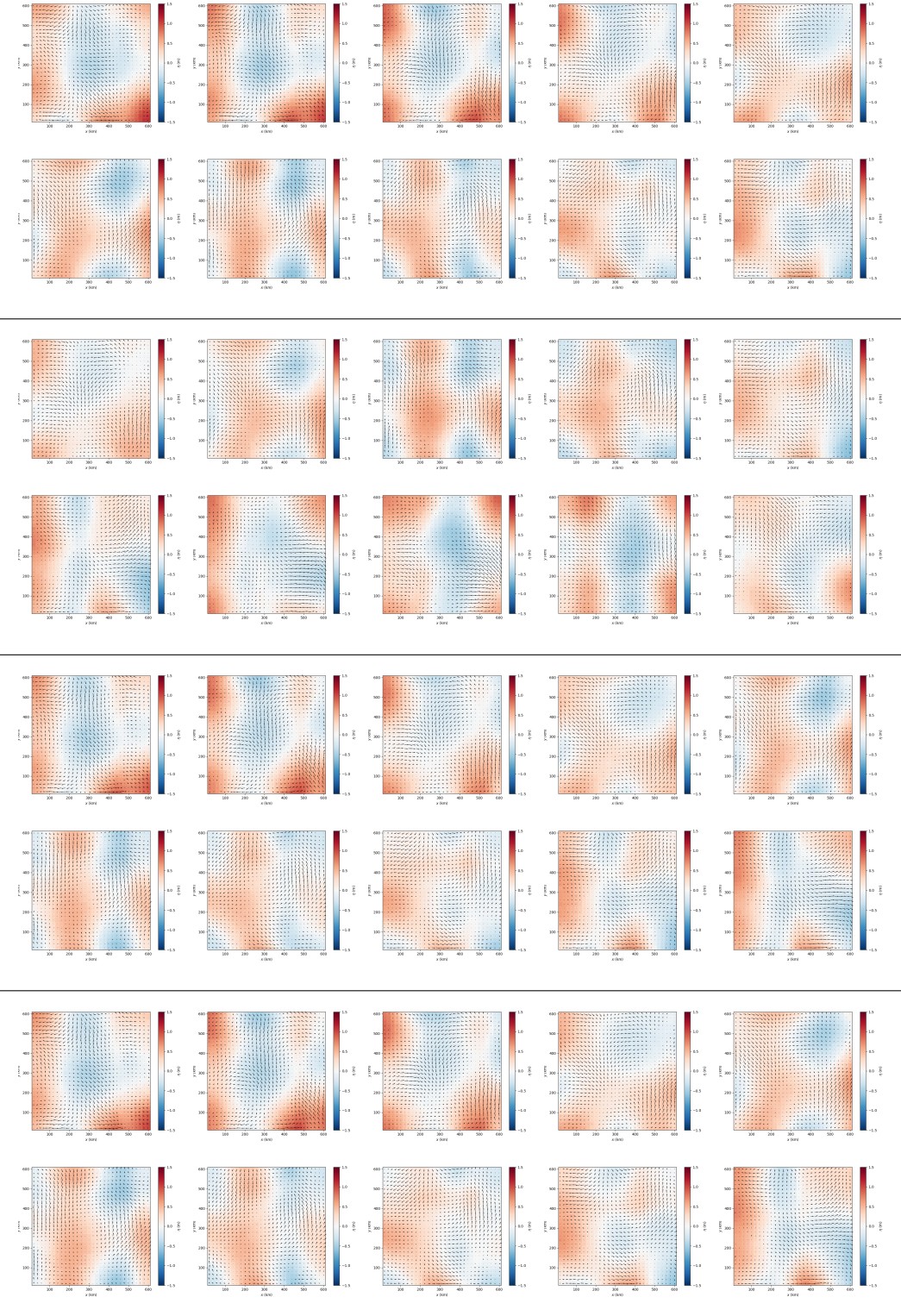

Figure 13: Examples of shallow water samples generated by models trained with PDE constraints. Each sample contains 50 frames and we uniformly visualize 10 of them.

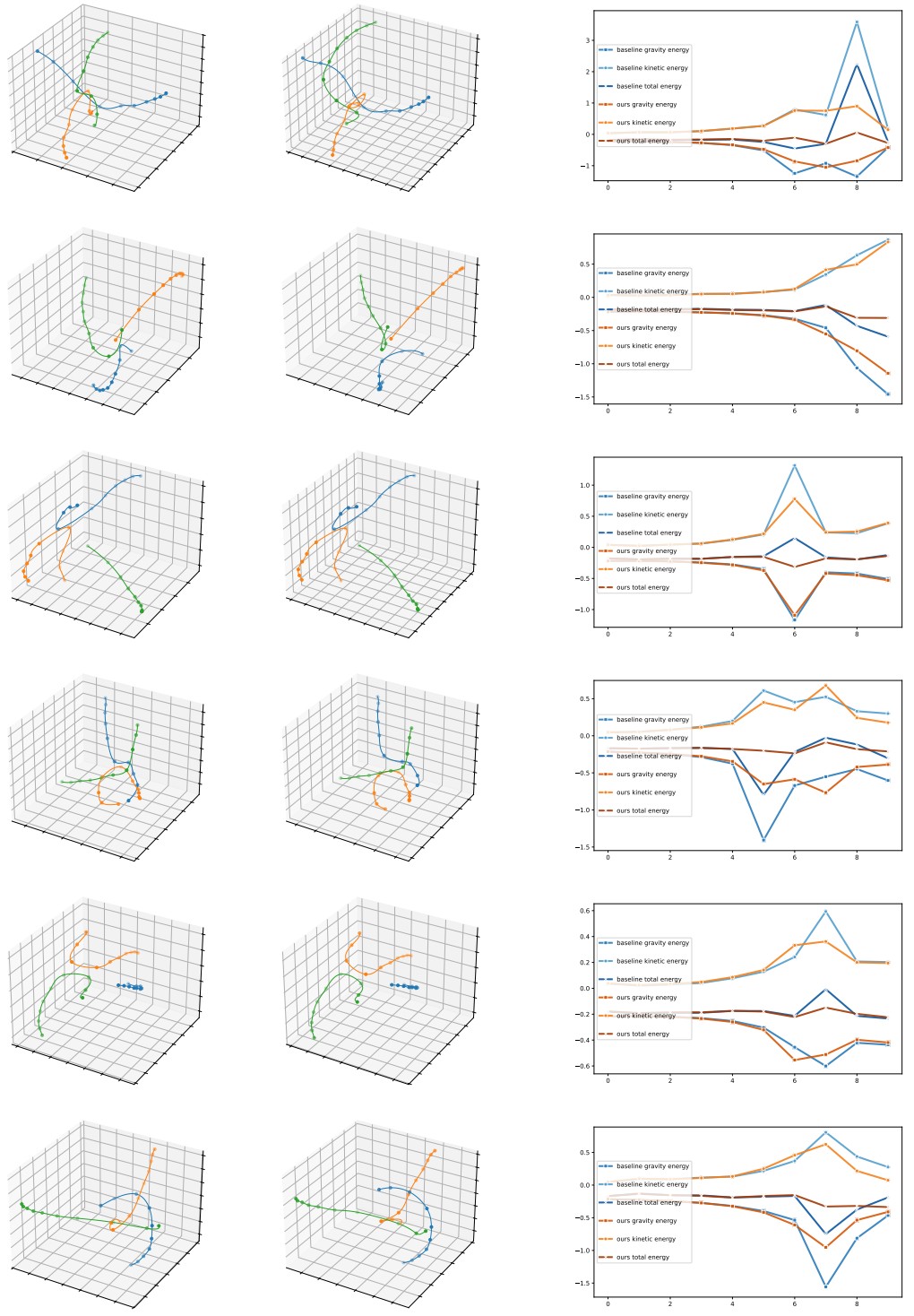

Figure 14: The figure illustrates examples of three-body samples produced by models trained with and without conservation constraints. The left panel represents the baseline model, the middle panel corresponds to our model, and the right panel depicts the energy as a function of time.

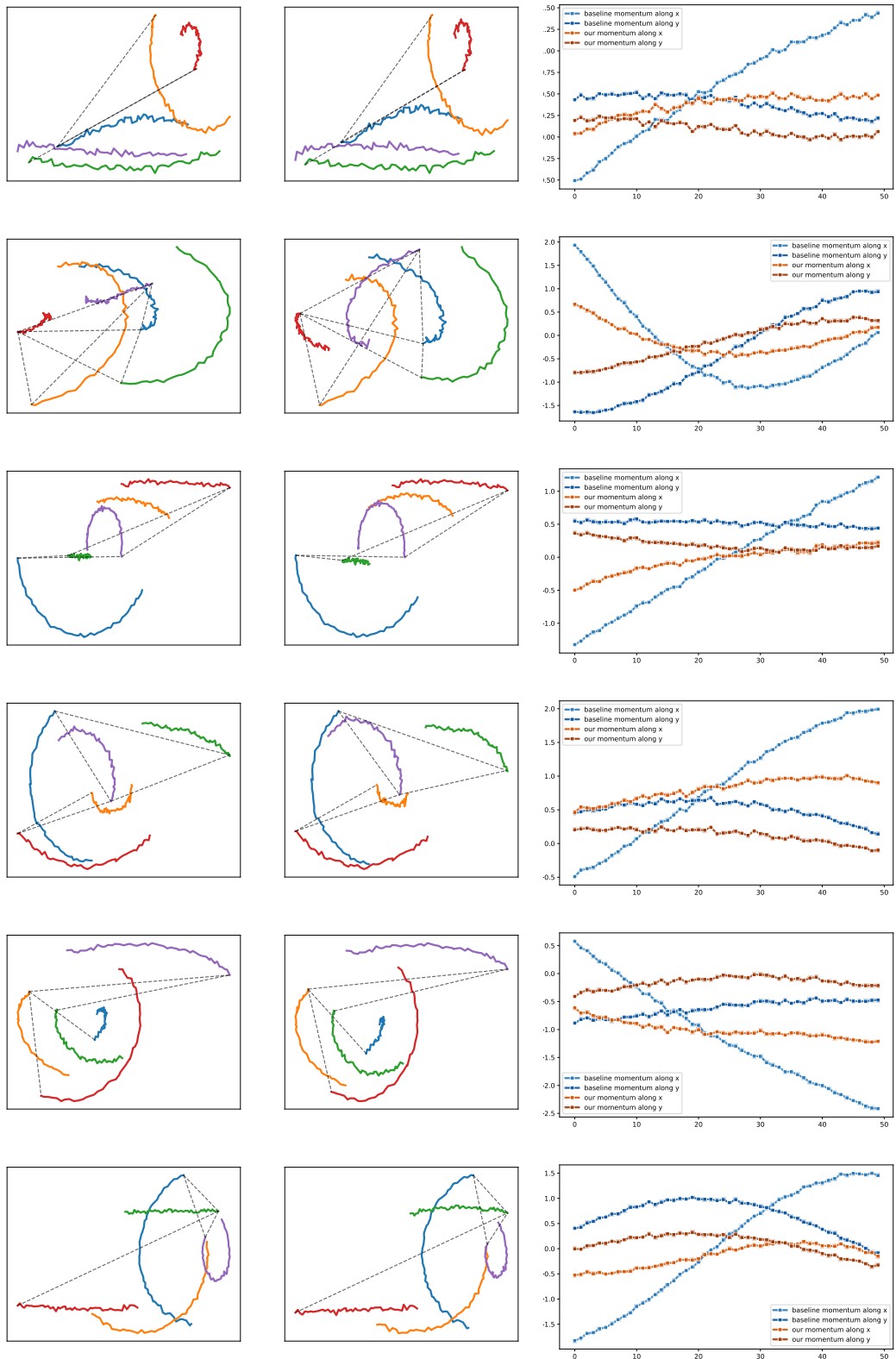

Figure 15: The figure illustrates examples of five-spring samples produced by models trained with and without conservation constraints. The left panel represents the baseline model, the middle panel corresponds to our model, and the right panel depicts the momentum as a function of time.

