# OpenReview forum: "Generating Physical Dynamics under Priors"
_ICLR.cc/2025/Conference — ICLR 2025 Poster_

### Official Review · Reviewer_2sjz · 2024-10-26

**Soundness:** 3
**Presentation:** 3
**Contribution:** 3
**Rating:** 6
**Confidence:** 3

**Summary:**

This paper proposes two ways to incorporate physical priors into generative models: (1) they inject distributional priors by choosing the proper equivariant models and (2) they incorporate physical feasibility priors by decomposing nonlinear constraints into elementary cases. The key is to identify elementary cases where Jensen's gap can be omitted and decompose complicated constraints into elementary cases. In general, this paper is a good contribution to the AI4Science community, where many scientific priors are available and should be incorporated into learning systems.

**Strengths:**

* The motivation is clear - this paper is a clear contribution to the AI4Science (esp AI4Physics) community where data-driven models should be combined with scientific inductive biases. Although physics-informed learning has been mainstream in scientific machine learning, physics-informed learning in the context of generative modeling is relatively new.
* This paper is both theoretically (supported by theorems) and practically sound (supported by experiments). The idea of decomposing a complex constraint into elementary cases is a good strategy.

**Weaknesses:**

* Empirical improvement is marginal (Table 1, 2, 3).
* This framework is useful when (1) we don't understand the system fully but (2) we know some partial information (existence of energy etc). However, all the examples in the paper are synthetic, so we know both the underlying equation and energy but "pretend" that we don't know about the equation. I understand this is only a proof of concept, but a more realistic example would greatly strengthen the paper.

**Questions:**

* Why does performance improve only incrementally after adding priors?
* In Equation (4) and line 205, are there typos? \nabla x -> \nabla_x?

---

> ### Author Response · Authors · 2024-11-14
>
> We thank reviewer 2sjz for identifying our contribution and insightful comments.
>
> # About marginal improvement of PDE datasets
> The incremental improvement in performance after incorporating priors is primarily due to the nature of the penalty loss term we introduced. As a regularization technique, the penalty loss enforces the prior gradually, constraining the model's parameters over time. This results in a more incremental enhancement of model performance, as opposed to a sharp improvement. The effectiveness of this approach depends on the interplay between the data, the model's capacity, and the strength of the penalty, which explains the observed gradual performance gain.
>
> While the improvement in performance may appear modest numerically, we have conducted multiple experiments and included error bars to account for variability (see general response section). The results consistently show that the observed improvement is indeed due to the enhancements in our approach, rather than random fluctuations. This suggests that the method is making a meaningful contribution, even if the magnitude of the improvement is small.
>
> # About application on the real-world dataset
> We appreciate your valuable feedback. While real-world dynamic datasets are highly relevant, they are often resource-intensive and costly to obtain, particularly in the form of densely sampled data. For this study, we have focused on datasets with dense sampling to demonstrate the effectiveness of our method. We view the application to real-world datasets as an important direction for future work, and we hope to explore this as more practical datasets become accessible.
>
> # About typos
> Thank you for pointing out the typo errors. We will carefully correct all the mentioned and other typos in the final version of the manuscript.

---

> > ### Comment · Reviewer_2sjz · 2024-11-20
> >
> > I'd like to thank the authors for the kind response and look forward to future work on real-world problems. I'll keep my score as 6.

---

> > > ### Author Response · Authors · 2024-11-20
> > >
> > > Thank you for reviewing our response and providing valuable feedback. We greatly appreciate your constructive comments and your acknowledgment of our contributions to AI4Science. We are delighted that our clarifications addressed your concerns, particularly regarding the performance on PDE datasets. Including error bars to demonstrate the effectiveness of our proposed methods was an important addition, and we are glad it met your expectations.

---

### Official Review · Reviewer_JiqC · 2024-10-28

**Soundness:** 2
**Presentation:** 2
**Contribution:** 2
**Rating:** 5
**Confidence:** 3

**Summary:**

This work explores ways of incorporating physics-informed priors in diffusion models for dynamical systems. In particular, they investigate **distributional priors**, aiming to incorporate appropriate symmetries (such as roto-translational invariance, permutation invariance, etc.), and **physical feasibility priors** as a way to encourage adherence to physical laws (e.g. momentum and energy conservation, PDE constraints, etc.). The manner in which the latter are imposed depends on the nature of the constraints, and the authors analyse several such cases separately (e.g. linear, multilinear, nonlinear, etc.).

The work provides empirical evidence on i) four PDE datasets, where PDE constraints are used, and ii) two particle dynamics datasets, where the diffusion model leverages the relevant physical laws, and training involves data augmentation to incorporate appropriate symmetries.

**Strengths:**

1. **Relevant topic.** There is increased interest in the dynamical system modelling community to incorporate relevant priors into models. Not only would this help with more efficient training, but perhaps more importantly, it should lead to better stability and capabilities to generalise out-of-distribution (OOD - when the test data comes from a different distribution than the training data).
2. **Wide range of datasets.** The experiments are performed on a wide range of datasets - four PDE ones, and two particle dynamics datasets. This helps at showcasing how the techniques would be employed in various scenarios, studying both linear and multilinear PDE constraints in the PDE datasets, as well as (reducible and general) nonlinear ones in the particle dynamics datasets.

**Weaknesses:**

1. **Contribution in distributional prior incorporation.** I believe there is some related work that is not being mentioned in section 3.1, as there are now plenty of works studying the incorporation of relevant priors in diffusion models. For example, Mathieu et al. [1] incorporate a series of geometric priors in infinite-dimensional modelling, also focusing on conditions for the diffusion process to be G-invariant assuming that the score network is G-equivariant.
2. **Lack of empirical evidence from the first methods subsection.** I do not think that the theory behind the first Methods subsection (Incorporating distributional priors) is sufficiently evidenced empirically.
- In the PDE datasets, the only result used empirically is that the data matching objective should be used, but this is rather weakly supported theoretically with the smoothness argument (although evidenced empirically in Table 4).
- In the particle dynamics datasets, I understand that finding the right architecture that incorporates all the relevant inductive biases can be hard, but if in the end data augmentation was used, how is the theory supported empirically in this case? I think you should include a discussion on this, as well as more details on how data augmentation was performed.
- Also related to the particle dynamics dataset, it seems that for the five-spring dataset you are using the EGNN architecture, (but you do not cite it - I am assuming you are referring to Satorras et al. [2]). I think this should be stressed in the main text with a brief discussion on the choice of architecture (referring to the inductive biases it incorporates). It's an important component given the chosen topic and I do not understand why it wasn't more clearly emphasised. And why are you using it in the five-spring dataset, but not in the three-body one?
- Finally, the section on ECM from Methods is not referenced later on in the paper. From my understanding, its purpose is to justify the data augmentation approach (in which case this connects to the previous point). I think a clearer connection between methods and how they are applied should be made.
3. **Lack of error bars.** The results do not contain any sort of error bars, which actually makes it very difficult to assess the effectiveness of the proposed methods. Especially in the PDE datasets results, the differences between the mean RMSEs are fairly small, and, depending on the errors, the distinction between the method w/o and w/ prior might actually turn out not to be significant. Could you please include them?
4. **Inadequate sample quality.** For the Darcy flow samples, the quality is very poor. I understand that no super-resolution or denoising procedures have been employed, but it seems like there is too much noise left in the samples. Do you take a final denoising step after the end of the diffusion process (by applying Tweedie’s formula)? Do you normalise the data? I can see that the data values for Darcy flow (with poor quality) are much lower (0.025-0.150) vs. Burgers (0.0-1.0), but the last sigma value is probably the same, so the proportional influence of the noise is higher in the Darcy flow dataset, potentially leading to poorer quality.
5. **Effectiveness of general nonlinear constraints.** The results on general nonlinear constraints are not that promising. In the three-body dataset, the results for noise matching + conservation of energy (general nonlinear) are very much comparable (especially for certain hyperparameters) to plain noise matching (although this is hard to assess because of the lack of error bars) - see Table 8. In the five-spring dataset (Table 9), the improvement is once again marginal for general nonlinear. Therefore, although I liked the idea of utilising the same hidden states as the model, I would argue that it did not prove to be effective in these experiments. If the authors agree, I think this limitation should be highlighted properly in the paper.
6. **No discussion on extra computational cost.** The paper does not discuss anything about the extra computational costs of the proposed method.

**Minor**

7. **Lack of thorough details/references on architecture.** In Table 7, I believe the Karras Unet should be referenced (from the EDM [3] paper). The paper also mentions that in the five-spring dataset the EGNN architecture is used but no reference is given (Satorras et al. [2]) and maybe the acronym should be defined for completeness.
8. **Normalised metrics.** For easier comparison between datasets, it might be better to use normalised metrics rather than absolute ones.
9. **Small legend in graphs and typos.** The font of the legends in the plots is too small (see Figure 2, Figure 8, Figure 9, etc.). There are also several typos throughout the paper (L107 extra “of”, L114 should be $\alpha_T$ and $\sigma_T$?, L271 related to the absolute, etc.).
10. **Inconsistent notation for score.** Sometimes you specify the score as $\nabla_{\mathbf{x}} \log q_t(\mathbf{x})$, sometimes as $\nabla_{\mathbf{x}} \log q_t(\mathbf{x}_t)$.

Overall, I think that the paper contains some nice ideas, but fails to clearly connect them to the empirical evidence, the experiments are poorly presented (with lack of error bars) and the effectiveness of some cases seems overstated.

[1] Mathieu, E., Dutordoir, V., Hutchinson, M.J., Bortoli, V.D., Teh, Y.W., & Turner, R.E. (2023). Geometric Neural Diffusion Processes. ArXiv, abs/2307.05431.

[2] Satorras, V.G., Hoogeboom, E., & Welling, M. (2021). E(n) Equivariant Graph Neural Networks. ArXiv, abs/2102.09844.

[3] Karras, T., Aittala, M., Aila, T., & Laine, S. (2022). Elucidating the Design Space of Diffusion-Based Generative Models. ArXiv, abs/2206.00364.

**Questions:**

1. **Related to W1** - What is the element of novelty in your exposition of incorporating distributional priors? If there is not any, this is not a problem, but then the relevant related works should be cited and the paper should indicate this clearly.
2. **Related to W3** - Can you also provide error bars on all results?
3. **Related to W6** - Could you comment on the extra cost of introducing the physical constraints using finite difference methods?
4. **Alternative objective** - For the distributional prior, have you ever tried a different objective that is a mix between the noise and data objective (for example, v-prediction [Salimans et al. [4]])?
5. For the PDE datasets, why don’t you provide the RMSE of the PDE constraints on all datasets?
6. From my understanding, you only impose either the momentum conservation constrain or the energy conservation constraint. What if in the particle dynamics datasets you imposed both a conservation of momentum and a conservation of energy penalty? Would this lead to instabilities?
7. When employing finite difference methods to approximate the differential equations, how do you make sure that the spatio-temporal discretisation of the datasets is fine enough to yield significant results? Is the discretisation the reason why the RMSE of the PDE constraints in Table 2 is not closer to 0?
8. Could you please include in the captions of Table 4 and 5 the quantity that you are reporting (I am assuming Traj error)?
9. Isn’t the result in Table 4 for Shallow Water with distributional prior the same as the result in Table 2 Shallow water? If so, shouldn’t it be 8.151 instead of 8.150? And for the three-body dataset, shouldn’t it be 2.5613 as in Table 3 instead of 2.6084, or is there a difference between those settings?

[4] Salimans, T., & Ho, J. (2022). Progressive distillation for fast sampling of diffusion models. arXiv preprint arXiv:2202.00512.

---

> ### Author Response · Authors · 2024-11-14
>
> We are grateful for the reviewer JiqC’s constructive feedback.
>
> # Comparison between related works on distributional priors and ours
> While we have listed two related works on distributional priors in Theorem 10 and 11 (line 1298-1317), we thank reviewer JiqC for bringing us to notice more related work of incorporating distributional priors in diffusion generative models. We will add more discussion about these works and add these contents to the related work section.
>
> ## Novelty of our theorem against Mathieu et al. [1]'s
> We believe that the theorem proposed by Mathieu et al. (2023) [1] is very similar to our cited theorem of Yim et al. (2023) [2] (in our paper, Theorem 11 at line 1308). We summarize the difference between our work and theirs in the following table. We can see that the conditions of the theorems and the conclusions are different.
>
> | | Conditions               | Property of the Score Function                      | Proof Tool                         |
> |-| -------------------------|---------------------------------------------------|------------------------------------|
> | Mathieu et al. (2023) [1] | The mean and covariance of SDE are suitably G-equivariant | G-equivariant                         | probability flow ODE/SDE via Fokker-Planck |
> | Yim et al. (2023) [2]    |   The mean and covariance of SDE are G-equivariant and satisfy certain relations                                          |    G-equivariant                           | Fokker-Planck/divergence theorem  |                                    |
> | Ours                 | G is volume-preserving diffeomorphism, isometry, and homothety | $\left(\mathcal{G}, \nabla^{-1}\right)$-equivariant | Chain rule                        |                                    |
>
>
> [2] Jason Yim, Brian L Trippe, Valentin De Bortoli, Emile Mathieu, Arnaud Doucet, Regina Barzilay, and Tommi Jaakkola. Se (3) diffusion model with application to protein backbone generation. arXiv preprint arXiv:2302.02277, 2023
>
> ## More theoretical results on the invariant distribution against Theorems in Mathieu et al. (2023) [1]
> In contrast to the related works discussed in the section "Equivalence Class Manifold for Invariant Distributions" (line 179) and Theorem 2, we also demonstrate that a $\left(\mathcal{G}, \nabla^{-1}\right)$-equivariant model that can predict the score functions in ECM, capable of predicting the score functions on the ECM, will also predict the score functions for all other points that are closed under the group operation. This result is novel.
>
> # About the evidence of distributional priors
> We conduct new ablation studies on distributional priors. Please refer to the general response section which supports our theoretical findings.
>
> # About error bars
> Thank you for your valuable feedback. In response to your comment, we have now included error bars in our experiment to better quantify the variability. The results can be seen in the general response section. The model's performance remains stable, and the improvements are primarily due to our proposed penalty loss, rather than experimental randomness. We believe this addresses your concern regarding the significance of the observed differences in the PDE dataset results.
>
> # About the detailed implementation of the backbone
> We apologize for not citing EGNN in the initial version of our manuscript; we will ensure the citations of all used models are included in the final version.
>
> For the implementation of the backbone, we have uploaded the codes and we will also include a more detailed explanation in the final version.
>
> Regarding your question about the choice of dataset, we did not apply EGNN to the three-body dataset because of its specific characteristics. The three-body dataset is a regular dataset, where each sample has an equal number of nodes, and the interactions between the nodes (representing the bodies) always exist due to gravitational forces. This makes the dataset well-suited for methods that directly capture temporal dependencies, such as GRU, without the need for graph-based approaches. Thus, we used GRU to extract the node features, as it effectively handles the time-dependent nature of the problem.
>
> On the other hand, the five-spring dataset has a more complex graph structure, where the spring connectivity between nodes significantly influences the dynamics. Since the graph structure plays an essential role in determining the system's behavior, we first use EGNN to capture the graph-based dependencies and extract node features. These features are then input into a GRU model to further capture the dynamics of the system. This combination allows us to leverage both the graph structure and temporal dependencies, which is crucial for modeling the five-spring system accurately.
>
> In the final version of the paper, we will also include a more detailed explanation of our implementation for each dataset to clarify our choice of models.

---

> ### Author Response · Authors · 2024-11-14
>
> # About sample quality
> We appreciate the reviewer’s concern regarding the quality of the generated samples. While it is true that some noise appears in our visualized samples, we would like to emphasize that the overall quality is still comparable to the samples found in existing predictive tasks on PDE datasets. The numerical comparison in Tab. 10 in Appendix E.3 demonstrates that our model is comparable against commonly used models in the computation of the PDE such as FNO [3], Unet [4], and  PINN [5].
>
> The main challenge in PDE datasets lies in accurately predicting the time evolution, which is inherently complex. However, our generative model has shown a strong ability to capture this dynamic behavior. For instance, in Fig. 10, although the generated sample contains some noise, it clearly exhibits recognizable patterns. In contrast, when comparing the heat plot of the PDE error between the generated and ground truth samples, the error pattern appears as random noise, with no discernible structure. This suggests that, despite the noise, the model is successfully predicting the underlying patterns of the system.
>
> To further support our argument, we provide additional visualizations of the conditional generative images alongside the corresponding ground truth samples in the updated PDF in Fig.12. These comparisons reinforce our belief that the model captures the key features and evolution dynamics of the PDE task, even though some noise is inevitably present in the generated samples.
>
> We believe that the observed noise arises from the limited capacity of the backbone model. When we apply the same backbone to simpler tasks, such as MNIST generation, it produces samples with minimal noise, provided that we use the same training and sampling strategy. Despite our careful tuning of the training process for this specific task, the generated samples still exhibit some noise, which we attribute to the backbone's current limitations in handling the complexity of the PDE dataset.
>
> We hope this clarification addresses the reviewer’s concerns, and we are confident that our generative model is effective in predicting the desired patterns despite the noise observed in some visualizations.
>
> [3] Zongyi Li, Nikola Kovachki, Kamyar Azizzadenesheli, Burigede Liu, Kaushik Bhattacharya, Andrew Stuart, and Anima Anandkumar. Fourier neural operator for parametric partial differential
> equations. arXiv preprint arXiv:2010.08895, 2020.
>
> [4] Olaf Ronneberger, Philipp Fischer, and Thomas Brox. U-net: Convolutional networks for biomedical image segmentation. In Medical image computing and computer-assisted intervention–
> MICCAI 2015: 18th international conference, Munich, Germany, October 5-9, 2015, proceedings, part III 18, pp. 234–241. Springer, 2015.
>
> [5] Maziar Raissi, Paris Perdikaris, and George E Karniadakis. Physics-informed neural networks: A
> deep learning framework for solving forward and inverse problems involving nonlinear partial
> differential equations. Journal of Computational physics, 378:686–707, 2019.
>
> # About the effectiveness of general nonlinear case
> Thank you for your valuable feedback. We agree that the results for general nonlinear constraints are only marginally improved. This is, in part, because we only add the constraints in the **model's hidden state**, which does not directly constrain model output as in other cases. However, in situations where the use of multilinear functions is not feasible, we resort to the general nonlinear method as a last resort. Therefore, the method we present for general nonlinear constraints is not intended as a primary solution, but rather as a fallback when other techniques are not applicable. Our approach primarily focuses on leveraging multilinear functions to simplify these nonlinearities into elementary cases, which we have successfully applied to scenarios involving the conservation of momentum and energy.
>
> We recognize that the general nonlinear case poses significant challenges, and while this work introduces the concept of incorporating physics-based feasibility priors into the diffusion model framework, it does not yet offer a universal solution for all cases. We acknowledge this limitation and intend to explore more effective strategies for handling general nonlinear constraints in future work. In the final version of the paper, we will expand on the marginal improvement observed for the general nonlinear case and clarify that it is a direction for future research.

---

> ### Author Response · Authors · 2024-11-14
>
> # About the extra computational cost
> The proposed method introduces a penalty loss to constrain the model, which incurs additional computational cost primarily from calculating the penalty loss function. We use the finite difference method to approximate the derivative, and the cost of this approximation is negligible compared to the forward pass cost of the chosen backbone models. Hence, the training cost for a minibatch is nearly identical whether using the proposed method or not. The time for inference is exactly the same.
>
> # Minors
> We will include detailed information about the implementation of the architectures, correct the typos you mentioned, and double-check the citations. Thank you for pointing these out. For line 114, "By choosing $\alpha_t \rightarrow 0$ and  $\sigma_t \rightarrow 1$", we mean that taking $\alpha_t \rightarrow 0$ and $\sigma_t \rightarrow 1$ along with $t \rightarrow T$. This is not a typo.
>
> # Response to questions
> ## About distribution priors
> We provide a detailed analysis in the paragraph above: Comparison between related works on distributional priors and ours. Also, **note that the theorems about invariant distribution in your provided reference [1] is built on Yim et al. (2023) [2], which we have already cited and discussed in Theorem 11 (at line 1308)**. Although we address a similar problem, our hypothesis and conclusions differ. Please refer to that section for further details.
>
> [2] Jason Yim, Brian L Trippe, Valentin De Bortoli, Emile Mathieu, Arnaud Doucet, Regina Barzilay, and Tommi Jaakkola. Se (3) diffusion model with application to protein backbone generation. arXiv preprint arXiv:2302.02277, 2023
>
> ## About velocity matching
> We tested the velocity matching method on the three-body dataset and encountered significant instability. Initially, we found that directly applying the velocity matching technique to match the velocity resulted in no decrease in the loss function. To address this, we attempted to scale the velocity matching objective and adjust the output during the reverse diffusion process. This change allowed the loss to decrease initially, but it began to grow again shortly thereafter.
>
> Next, we reduced the initial learning rate from 1e-3 to 1e-5 and retrained the model. Despite this adjustment, the model still failed to generate meaningful samples on the three-body dataset. In contrast, when we applied the same framework to the MNIST dataset and trained for just 16 epochs, the model produced high-quality samples.
>
> Based on these results, we believe our implementation is correct, but the velocity matching method may be inherently more difficult to apply to the three-body dataset, likely due to its more complex dynamics.
>
> ## About the metric of the PDE dataset
> In the previous work [6], authors usually do not normalize metrics over datasets. In our work, we use RMSE as the evaluation metric for conditional generation tasks where there is a single ground truth. For unconditional generation tasks, we use the PDE feasibility error as the metric, since a widely recognized model for calculating the FID metric is not available.
>
> [6] Makoto Takamoto, Timothy Praditia, Raphael Leiteritz, Daniel MacKinlay, Francesco Alesiani,
> Dirk Pfluger, and Mathias Niepert. Pdebench: An extensive benchmark for scientific machine ¨
> learning. Advances in Neural Information Processing Systems, 35:1596–1611, 2022.
>
> ## About combining multiple penalty loss
> The experimental results show that combining multiple penalty losses maintains stability. However, the improvement achieved is only marginal compared to using a single penalty loss.
>
> ## About using finite difference for derivative approximation
> We input samples in the dataset into the function calculating the physics feasibility error and found that the error is very small (for example, advection: 0.02, burger: 0.15, shallow water: 1e-17) compared with the generated samples.
>
> ## About Tab. 4 and 5
> In Table 4, the metric for the three-body dataset is the average of the trajectory error and velocity error. For the spring dataset, we report the dynamic error. We apologize for any confusion and will clarify the metric setting.
>
> For all metrics, we apply rounding down. Therefore, 8.1506 in Table 2 is rounded to 8.150 in Table 4. For the three-body metric, we calculate the mean of the trajectory and velocity errors in Table 4, so 0.5 * (2.5613 + 2.6555) in Table 3 equals 2.6084 in Table 4. We apologize for the confusion and will ensure this is clearly explained in the final version.

---

> ### Comment · Reviewer_JiqC · 2024-11-17
>
> Thank you for your detailed response, clarifications, and the additional experiments. I have a few follow-up comments:
>
> **Error Bars**
>
> The RMSE results for the PDE constraints, along with the small variability, are expected given the physical feasibility priors. However, I am more interested in the forecasting examples—Advection and Darcy Flow. For Advection, the results are within 1 unit of error, but for Darcy Flow, you did not provide error metrics. Could you clarify why these metrics are missing?
>
> **Sample Quality**
>
> I remain concerned about the setup of the Darcy Flow experiment, specifically in relation to the diffusion process, which appears to introduce excessive noise in the predicted samples. Was the data normalised before training the diffusion model, or were the samples used within the original range (as shown in Figure 4)?
> - *The numerical comparison in Tab. 10 in Appendix E.3 demonstrates that our model is comparable against commonly used models in the computation of the PDE, such as FNO [3], UNet [4], and PINN [5].* To fully understand the comparison, it would be helpful to include an estimate of the errors and clarify the data range. If the original data lies predominantly in the range $0.01–0.1$, then an RMSE of $2.261 \times 10^{-2}$ is not negligible.
> - *This suggests that, despite the noise, the model is successfully predicting the underlying patterns of the system.* I interpret this to mean the model effectively captures low-frequency information but struggles with high-frequency details. In forecasting tasks, particularly for highly nonlinear systems, this limitation can propagate and lead to significant errors during rollouts [1], so I don't think it should be overlooked.
> - *Still exhibit some noise, which we attribute to the backbone's current limitations in handling the complexity of the PDE dataset.* This noise is less pronounced in other PDE datasets, hence my comment about a potential issue in the setup of the Darcy Flow experiment. Revisiting this experiment to ensure proper normalisation and setup would make your results more conclusive.
>
> **Effectiveness in the General Nonlinear Case**
> This is perfectly fine, as long as you clearly state it in the paper.
>
> **Velocity Matching**
>  This is interesting, but maybe not completely unexpected given the difference between the characteristics of PDE states vs natural images (where v-prediction proved successful). Do you believe you might see more stable results using the formulation in [2]?
>
> [1] Lippe, P., Veeling, B.S., Perdikaris, P., Turner, R.E., & Brandstetter, J. (2023). PDE-Refiner: Achieving Accurate Long Rollouts with Neural PDE Solvers.
>
> [2] Karras, T., Aittala, M., Aila, T., & Laine, S. (2022). Elucidating the Design Space of Diffusion-Based Generative Models.

---

> > ### Author Response · Authors · 2024-11-17
> >
> > Thank you for your feedback.
> >
> > # Error bar of Darcy Flow
> > The error metrics for Darcy Flow are currently missing because the model has not yet fully converged, and we are still in the training process for this experiment. As soon as the model reaches convergence, we will be able to provide the necessary error metrics. (Also for the five-spring dataset.)
> >
> > # More on Darcy Flow
> > ## About data normalization
> > The predicted values in the Darcy Flow dataset range from 0 to 1.25. Approximately 6% of the samples contain pixels with values exceeding 0.7. Given this, we chose not to apply data normalization. Additionally, we directly utilized the data loader from [1], where normalization was also not applied either. For clarity, we have included a visualization of the pixel distribution in the updated version of our paper in Fig. 17.
> >
> > [1] Wei P, Liu M, Cen J, et al. PDENNEval: A Comprehensive Evaluation of Neural Network Methods for Solving PDEs[J].
> >
> > ## About relative error
> > We are fully aware that reviewer JiqC wants to help readers have a better understanding of the precision of the prediction. As can be seen in Fig. 4 and in the updated Fig. 17, each sample of the Darcy Flow dataset has different value ranges. Hence, we cannot quantify the relative error using the RMSE metric even with the provided value range of the whole dataset. However, in parallel to your direct concern, we can provide the mean of relative error over all tested samples whose formula of each sample is given as:
> >
> > $$\frac{mean( | \text{ground truth}_{ij} - \text{generated} _{ij} |)}{\max(\text{ground truth} _{ij}) + 1e-6}$$
> >
> > where max is taken over all pixels. The results show that our method achieves only 3.43% relative errors.
> >
> > ## More on noise in Darcy Flow
> > ### Noise removal experiment
> > In response to the reviewer's suggestion, we conducted additional experiments using traditional methods to reduce the noise in the generated samples of Darcy Flow. Specifically, we applied the median_filter in scipy.ndimage in Python, with a filter size of 8, to smooth the images. The results of this experiment are presented in Figure 12 of the updated paper. Our findings show that using this noise reduction method, the RMSE for the model with prior decreased from 2.174 to 2.012, and the RMSE for the model without prior decreased from 2.261 to 2.123.
> >
> > We still want to clarify that the primary focus of this paper is on incorporating the prior to improve the model fit of the score function. As such, we have not discussed other potential approaches for improving the sample quality, such as more advanced denoising techniques or the use of specialized models like FNO, which are designed for PDE datasets and inherently produce smoother samples by suppressing high-frequency components. We believe designing improved backbones can be our future work.
> >
> > ### About high-frequency noise
> > We thank the reviewer for highlighting the importance of addressing high-frequency noise, and we add a new discussion on this issue in the revised paper in Appendix E. 6. Additionally, when comparing the Darcy Flow dataset with other PDE datasets, it is important to note that the Darcy Flow dataset includes the second derivative of the PDE, while other datasets do not. This may explain the presence of noise in the generated samples. We have not yet proposed a specific method for handling the second-order derivative, as it is not the primary focus of this paper.
> >
> > # About velocity matching
> > We are using the velocity matching approach as described in [2]. In that paper, the authors demonstrate the equivalence between noise, data, and velocity matching in Theorem B.1. By applying the same reasoning used for the equivalence between noise and data matchings, it becomes clear that the optimal values of the velocity matching objective do not possess the same equivalence or smoothness properties. This lack of equivalence and smoothness may explain why the current backbones are unable to effectively utilize velocity matching on the particle dynamics dataset.
> >
> > [2] Kaiwen Zheng, Cheng Lu, Jianfei Chen, and Jun Zhu. Improved techniques for maximum likelihood estimation for diffusion odes. In International Conference on Machine Learning, pp. 42363–42389. PMLR, 2023

---

> > > ### Comment · Reviewer_JiqC · 2024-11-18
> > >
> > > *“The error metrics for Darcy Flow are currently missing because the model has not yet fully converged, and we are still in the training process for this experiment.”* Does this imply that the original metrics reported in Table 1 correspond to a model that has not converged and you will modify the results? If this is the case, that’s fine, at least it might explain the excessive noise in the samples, but I would also suggest not rushing experiments up. I believe there was enough diversity in the experiments even without the Darcy Flow, so better to have some well trained models on a couple of experiments, rather than compromise on model quality for more experiments.
> > >
> > >
> > > *“Additionally, we directly utilized the data loader from [1], where normalization was also not applied either.”* I see, but the paper that you cite does not focus on any diffusion-based model. However, in your case, I see in **Appendix E - Diffusion details** that you are using the VPSDE, which maintains the variance of the noised up states to 1 **provided the variance of the original data is 1**, which is one reason people tend to normalise the dataset. But maybe this is not something crucial.
> > >
> > > **Noise removal** I appreciate the extra effort put into this experiment, but my main message was that if a diffusion model is trained well-enough, you shouldn’t see that in the samples, not that you need to do some post processing to remove the noise.
> > >
> > > *“the primary focus of this paper is on incorporating the prior to improve the model fit of the score function. As such, we have not discussed other potential approaches for improving the sample quality”*
> > > I understand this, and I wouldn’t want to depart the discussion from the main themes of the paper. However, the reason why I am insisting on the Darcy Flow performance is because an improvement (using the physical priors) over an unreliable baseline (without physical priors) does not allow me to accurately assess the effectiveness of incorporating the physical prior. It gives no indication about whether, starting from a reliable baseline, the same sort of improvement would be observed.
> > >
> > > Overall, I think the paper offers some valuable insights, but it could still be improved with more carefully designed experiments. As such, I increase my score to 5.

---

> > > > ### Author Response · Authors · 2024-11-19
> > > >
> > > > # Missing results in the rebuttal updates.
> > > > _“The error metrics for Darcy Flow are currently missing because the model has not yet fully converged, and we are still in the training process for this experiment.”_ To clarify, the results reported in the original paper correspond to models that have fully converged. We ensured convergence by stopping training only when the learning rate decayed to below 5e-7. Currently, we are still running additional experiments to include error bars in the results.
> > > >
> > > > # More on the quality of the Darcy Flow dataset.
> > > > We have revised the sampling process for the Darcy flow dataset. When it is near the end of the reverse diffusion process, we use the model's output of $E[x_0 | x_t]$ as the final generated sample as suggested by reviewer JiqC (_a final denoising step after the end of the diffusion process by applying Tweedie’s formula_). This adjustment significantly improves the visual sample quality for the Darcy flow dataset. However, for the other datasets, the results remain largely unchanged. The updated visualization of the generated samples can be found in Fig. 11 of the revised paper. The results can be seen in the following table.
> > > >
> > > > | Method      | Original sampling | New sampling |
> > > > |-------------|------------------------------|--------------------------------|
> > > > | w/o prior   | $2.261$              | $2.016$                          |
> > > > | w/ prior     | $2.174$              | $1.954$                          |
> > > >
> > > >
> > > > We would like to emphasize that **our methods are evaluated across six diverse datasets**, ranging from PDE dynamics to particle dynamics, which exhibit very different patterns. Notably, for the particle dynamics dataset, where the constraints are highly nonlinear and complex, incorporating priors using our methods significantly improves the sample quality.
> > > >
> > > >
> > > > If you have any further concerns regarding the Darcy flow dataset experiment, please feel free to reach out.

---

> > > > > ### Comment · Reviewer_JiqC · 2024-11-25
> > > > >
> > > > > Why do you need another model to converge to provide error bars? Can't you use the already-trained model to sample and provide the resulting error bars? Or by convergence, do you refer to something different than training convergence (which is what I was assuming)?
> > > > >
> > > > >
> > > > > In my experience, applying Tweedie's formula as a final denoising step does tend to lead to improved sample quality indeed and I am glad it was the case in here too.

---

> > > > > > ### Author Response · Authors · 2024-11-26
> > > > > >
> > > > > > Initially, we interpreted the request for error bars as requiring an assessment of variability stemming from both the training process (by varying the seed for training initialization) and the sampling process (by varying the seed for sample generation). This interpretation led us to retrain the denoising model multiple times with different random seeds, as we believed this would comprehensively capture the uncertainty inherent in both stages of the generative process. (In the other updated results, the reported error bars were calculated under the setting where both the training seed and the sampling seed were varied (3 or 5 times).)
> > > > > >
> > > > > > However, upon re-reading your comment and clarification, we now understand your suggestion to focus solely on the variability in sample quality due to changes in the sampling seed, using a single-trained model.
> > > > > >
> > > > > > To address this, the results can be seen in the following table:
> > > > > > | Method                   | Darcy flow |
> > > > > > |--------------------------|----------------------------------    |
> > > > > > | w/o prior            | 2.0648   $\pm$  0.0600           |
> > > > > > | w prior               | **1.9678 $\pm$  0.0651**          |

---

> > > > > > > ### Author Response · Authors · 2024-12-02
> > > > > > >
> > > > > > > Thank you again for your valuable feedback on our submission. We hope that the revisions we have made and the clarifications provided in our response have adequately addressed your concerns. If you have any further questions or if there are aspects that still need clarification, please do not hesitate to let us know. We look forward to your feedback.

---

### Official Review · Reviewer_RSW6 · 2024-10-31

**Soundness:** 3
**Presentation:** 3
**Contribution:** 3
**Rating:** 8
**Confidence:** 3

**Summary:**

The paper proposes a novel framework to generate physically feasible dynamics by incorporating physical priors into diffusion-based generative models. Unlike traditional generative approaches, which often disregard physical constraints, this model integrates two types of priors: distributional priors (such as invariance to roto-translation) and physical feasibility priors (e.g., conservation laws for energy and momentum, and partial differential equation constraints). By embedding these priors into the generative process, the method produces realistic dynamics across a variety of physical systems.

**Strengths:**

- The work is innovative in its focus on embedding physical feasibility directly into diffusion-based generative models, specifically through the use of both distributional and physical priors. By designing a process that enforces these constraints, the model is capable of generating realistic dynamics, distinguishing it from more traditional generative approaches that may ignore or only partially enforce physical laws.

- Moreover, the paper is well-written and is thorough with both the theoretical part and the empirical results for the proposed model.

- The paper addresses a challenge in generating accurate physical dynamics with AI methods. It enables more reliable and scientifically sound simulations in areas like environmental, material science, and possibly the fluid simulations.

**Weaknesses:**

- The experiments focus on synthetic datasets or physics-inspired datasets, and while this is valuable, additional testing on real-world, noisy datasets would definitely help.

- If possible, comparing the model's performance with more baselines will make the empirical results more convincing. Yet it might be difficult to find the methods from the exactly same field, but some methods from, e.g. time-series forecasting, can be taken into consideration.

**Questions:**

- The model includes various hyperparameters, especially in the weighting of physical feasibility losses. How sensitive is the model to these settings? Could the authors provide guidance on tuning these parameters, or propose default values based on their findings?

- What does "intrinsic structures of pairwise distance" refer to?

- Does the resolution matter for "reducible nonlinear cases"?

- As I noticed, all of the datasets in this work seem to be with multi-dimensional features. Yet in biology, we usually just have single-dimensional features possibly collected at a time-step. Will this affect the performance of the method?

- I think the five springs dataset is probably more like an ODE dataset instead of a PDE. The transition functions are second-order ODEs.

- Following last question, how about reporting the errors of NRI (Kipf et al., 2018) on the experiments in Section 4.2 as well?

- I have a tiny suggestion on the setup of the paper. If it is hard to fit a whole section of related works in the main body, it would be a good practice to merge a part of the related works to the section of preliminaries. Readers need the clear message in the related works to recall the research gap.

- Please check the score function in line 146, it does not consistent with the one in line 115. Please be careful about the suffices.

**Details Of Ethics Concerns:**

N.A.

---

> ### Author Response · Authors · 2024-11-14
>
> We are grateful for reviewer RSW6's thoughtful suggestions.
>
> # About application on the real-world dataset
> We acknowledge the importance of testing our approach on real-world datasets. However, real-world datasets involving dynamic systems are often expensive and challenging to acquire, particularly in the context of dense sampling. So far, we have focused on datasets with dense, well-sampled data to ensure the robustness of our approach. We plan to explore real-world applications in future work as more accessible datasets become available.
>
> # About more baselines
> We have included time-series forecasting results on PDE datasets in Tab. 10 of Appendix E.3. Specifically, we compared our models with FNO [1], Unet [2], and PINN [3]. In this section, we compare the performance of our generative models, specifically for generating Dirac distributions (prediction tasks), against commonly used models in physics dynamics prediction. While we did not fine-tune the architectures specifically for physics dynamics generation, our results still demonstrate competitive performance when compared to these models.
>
> # About the sensitivity of hyperparameters
> We have already presented the influence of hyperparameters of the backbone model in Tab. 8 and 9 in Appendix E.2. Regarding the hyperparameters of the penalty loss weight, our experimental results show that the model performance is not highly sensitive to variations in the loss weight. Specifically, we conducted a search for the loss weight across a logarithmic scale, testing it approximately five times. To further illustrate this, we provide an example of how the loss weight affects model performance on the three-body dataset with momentum conservation in the table below:
>
> | loss weight | traj     | vel      | energy   |
> |------|----------|----------|----------|
> | 0 (baseline) |  2.5613 |  2.6555 | 3.8941
> | 0.005   | 2.5691 | 2.6446 | 3.9429
> | 0.01     | 2.3497 | 2.4284 | 3.5943
> | 0.05     | 2.1409 | 2.2529 | 4.1116
> | 0.1       | 2.3432 | 2.4326 | 4.1938
> | 0.5       | 2.2146 | 2.3617 | 5.3853
> | 1.0       | 2.2461 | 2.3543 | 4.6903
>
> # About the intrinsic structure of pairwise distance
> We mean that coordinates that have undergone rotation and translation maintain their pairwise distances. We will clarify it in our final version.
>
> # About the resolution of images
> Our experiments of reducible nonlinear cases focused on particle dynamics datasets, which do not involve image data.
>
> The resolution typically impacts the accuracy of the numerical solution to the PDE datasets. A higher resolution allows for a more accurate approximation of the derivative term using the finite difference method. However, in our experiments, we tested the feasibility constraints on the given dataset and found that the numerical error is minimal. As a result, the resolution does not significantly affect the application of physics constraints. It is important to note, though, that if the resolution of the image representing the PDE solution is too low to effectively approximate the derivative using the finite difference method, the proposed approach would not be applicable in such cases.
>
> # About the application on single-dimensional datasets.
> Thank you for your insightful comment. The proposed method is indeed applicable to datasets with single-dimensional features as well. While the examples in this work focus on multi-dimensional feature sets, our approach is flexible and can be extended to handle single-dimensional data.
>
> However, regarding the incorporation of physics feasibility constraints in the 1D case, I am uncertain about how these constraints would manifest. In higher-dimensional feature spaces, the constraints might have more complex forms and thus a more significant impact on the optimization process. In the 1D case, if the constraints are trivial or simple, they may not significantly improve the model's performance.
>
> Could you provide an example of such biological datasets where only single-dimensional features are used? This would help us better understand the context.
>
> # About the five-spring dataset
> This dataset is indeed governed by an ODE, and we refer to it as a "particle dataset" to distinguish it from the PDE-based dataset. In the original dataset introduced by NRI (Kipf et al., 2018), the authors clamp the maximum spring force and the positions of the particles for numerical stability. However, when generating our dataset, we remove these constraints, as they violate energy and momentum conservation. Additionally, while NRI's approach focuses on prediction tasks, our work involves a generation task, which makes their method not directly applicable to our setting. We will, however, report the error using their predictive training scheme in our final work.

---

> > ### Author Response · Authors · 2024-11-14
> >
> > # About the related work
> > Thank you for your suggestion regarding the related works section. We will incorporate part of the related works into the preliminaries section in the final version, as you recommended, to ensure a clearer connection to the research gap for the readers.
> >
> > # About typos
> > We will correct all the mentioned typos and double-check our scripts again.
> >
> > [1] Zongyi Li, Nikola Kovachki, Kamyar Azizzadenesheli, Burigede Liu, Kaushik Bhattacharya, Andrew Stuart, and Anima Anandkumar. Fourier neural operator for parametric partial differential
> > equations. arXiv preprint arXiv:2010.08895, 2020.
> >
> > [2] Olaf Ronneberger, Philipp Fischer, and Thomas Brox. U-net: Convolutional networks for biomedical image segmentation. In Medical image computing and computer-assisted intervention–
> > MICCAI 2015: 18th international conference, Munich, Germany, October 5-9, 2015, proceedings, part III 18, pp. 234–241. Springer, 2015.
> >
> > [3] Maziar Raissi, Paris Perdikaris, and George E Karniadakis. Physics-informed neural networks: A
> > deep learning framework for solving forward and inverse problems involving nonlinear partial
> > differential equations. Journal of Computational physics, 378:686–707, 2019.

---

> > > ### Comment · Reviewer_RSW6 · 2024-11-22
> > >
> > > Dear Authors,
> > >
> > > Many thanks for the rebuttal and the additional results. As it might be too late to run more experiments on biological data, I have the following benchmark which contains the data you might be interested at: <Pratapa, Aditya, et al. "Benchmarking algorithms for gene regulatory network inference from single-cell transcriptomic data." Nature methods 17.2 (2020): 147-154.> It would be nice to see the experimental results on the datasets mentioned in the future work. Yet I am still curious about the reason of not running NRI on the ‘particle dataset’. Although the dataset used in NRI is different from the one in this work, NRI has no bias on the choice of the datasets and I believe it can also work on the proposed one. A general overview on the prediction error comparison on the dynamics would be sufficient.
> > >
> > > Warm regards.

---

> > > > ### Author Response · Authors · 2024-11-23
> > > >
> > > > # About the application of biological data
> > > > Thank you for sharing the reference and for your valuable suggestion. While we acknowledge the importance of benchmarking on a wider range of data, we have limited experience in the biological domain and need more time to understand the data and assure the correctness of our implementation. However, as suggested by you, we will make our effort accelerating the implementation and experiments before the rebuttal deadline. We will update the results once available. Again, we appreciate your understanding and valuable suggestions.
> > > >
> > > > # About NRI
> > > > Thank you for your comment on including the results of NRI on the five-spring dataset in the predictive training scheme. We have included it in the updated paper in Tab 11 (lines 1242-1258). We use the decoder of NRI, and show that the generative method is comparable to the predictive methods.
> > > > | Method                   | Dynamic error ($\times 10^{-3}$) | Momentum error | Energy error |
> > > > |--------------------------|----------------------------------|----------------|--------------|
> > > > | noise matching            | 2.5178                           | 5.3511         | 1.0891       |
> > > > | noise matching + priors   | **2.4329**                       | **0.3687**     | **0.7448**   |
> > > > | NRI predict               | 2.4471                           | 5.2234         | 1.4577       |
> > > >
> > > > Note that for other tables, we use the mean of error between the generated dynamics and the ground truth dynamics solved by finite element methods in the following 8 steps, while in this table, we use only 1 step.

---

> > > > > ### Comment · Reviewer_RSW6 · 2024-11-26
> > > > >
> > > > > Many thanks for the additional comparison with NRI. I appreciate your efforts on all of the additional experiments. I will raise my score to 8 to support the acceptance of the submission.

---

> > > > > > ### Author Response · Authors · 2024-11-27
> > > > > >
> > > > > > We sincerely thank reviewer RSW6 for thoughtful feedback and recognition of our work's novelty in embedding physical feasibility into diffusion-based generative models. We appreciate your acknowledgment of our approach's innovation, thorough theoretical grounding, and potential impact on generating reliable and scientifically sound simulations.

---

### Official Review · Reviewer_QDux · 2024-11-04

**Soundness:** 3
**Presentation:** 3
**Contribution:** 2
**Rating:** 6
**Confidence:** 2

**Summary:**

This work proposes a framework that incorporates physical priors, including distributional priors and physical feasibility priors, to generate physically realistic dynamics with diffusion-based models.

**Strengths:**

1. The manuscript is well-written and easy to follow.
2. Thorough derivations are provided in the main content and appendix.
3. Sufficient experiments and ablation studies are conducted to validate the effectiveness of the proposed method.

**Weaknesses:**

Please see the Question part.

**Questions:**

- The three contributions listed in Section 1 appear similar and convey the same meaning. Accurately summarizing the manuscript's contributions will be more helpful for the readers.

- Why distribution priors (such as translational rotational invariance) are important for generating physical dynamics? As stated in [1], the invariant sampling and invariant loss functions by restricting architecture designs often sacrifice empirical performances. Are simple data augmentations that approximate probability equivariance also available in physical dynamics generation? Experiments with data augmentations that approximate probability equivariance might help analyze the effectiveness of equivariant models.

- How are the physical priors selected in this work? It appears that the roto-translational invariance and the priors based on energy and momentum conservation laws are chosen arbitrarily. There seems to be no systematic analysis that explores all possible categories of physical priors or how to incorporate each type of prior into diffusion models. I believe there are many more physical priors that generative models should be constrained by. Categorizing them and discussing each category systematically would provide more insight.

- In Line 498-508, the paragraph title is "Data matching vs noise matching". However, the ablation study is conducted for incorporating a distributional prior or not, which is expected to investigate the training objective instead.

---
[1] SwinGNN: Rethinking Permutation Invariance in Diffusion Models for Graph Generation. TMLR 2024.

---

> ### Author Response · Authors · 2024-11-14
>
> We thank reviewer QDux for the comments.
>
> # Summarization of Sec. 3.1.
> Thank you for your feedback. We agree that the three contributions listed in Sec. 3.1 can be more clearly distinguished. The current summary in Remark 1 was intended to highlight these contributions, but we will revise this section in the final version to improve clarity and provide a more accurate and concise summary of the manuscript's contributions.
>
> # About invariant sampling
> ## Comparison between SwinGNN and ours
> In SwinGNN [1], authors argue that invariant models have more restrictive architectural designs. Thus, this may sacrifice the performances. We agree with this perspective, but we want to emphasize that our mathematical derivation shows that when the training objective is optimized, the trained model should be $\left(\mathcal{G}, \nabla^{-1}\right)$-equivariant. Given two models of equivalent capacity, where one is $\left(\mathcal{G}, \nabla^{-1}\right)$-equivariant and the other is not, we aim to select the former model. We will add more discussion about the possibility of sacrificing performance using the equivariant models in our final work.
>
> Also, in SwinGNN [1], the authors primarily focused on the task of graph generation, whereas our work targets the generation of dynamics. These are distinct tasks, and the significance of invariance may vary between them.
>
> ## Invariance is important in our datasets
> We have conducted additional ablation studies that provide valuable insights. Specifically, for particle dynamics datasets, our experiments show that invariance data augmentation improves the model's performance. These results are detailed in the general response section.
> We believe that the impact of invariance may be architecture-dependent. Many recent state-of-the-art models have leveraged transformers as their backbone, and it is possible that transformers naturally align with certain symmetries, such as permutational equivariance. However, as discussed in Appendix E.4, we also explored transformer-based architectures. Due to the Markov properties inherent in particle dynamics, we found that recurrent architectures performed better than transformers for this specific dataset.
>
> # Categories of physical priors
> In our work, we categorize physical priors into five types, as outlined in Sec. 3.2 as 1) linear cases, 2) multilinear cases, 3) convex cases, 4) reducible nonlinear cases, 5) general nonlinear cases. We evaluate the proposed methods across all of these categories using both PDE and particle dynamics datasets. The specific forms of the priors and their categorization are detailed in Tab. 6 in Appendix C.
>
> The momentum conservation laws were selected because they typically involve linear constraints, while the energy conservation laws were chosen due to their often nonlinear nature. For example, in the three-body dataset, the energy conservation involves terms like $1/r$ while in the five-spring dataset, it includes terms like  $r^2$. Also, we think that conservation laws can be used to measure the feasibility of the generated dynamics.
>
> # About the paragraph title of the ablation studies
> We will revise the paragraph title as suggested by the reviewer. Thanks for the advice.

---

> > ### Comment · Reviewer_QDux · 2024-11-25
> >
> > I appreciate the authors' thorough response and have checked reviews from other reviewers. All my concerns are solved. But I'd like to mention that I'm not an expert in physical sciences, so I keep my current score as "marginally above the acceptance threshold".

---

> > > ### Author Response · Authors · 2024-11-25
> > >
> > > Thank you for your feedback and for reviewing our responses. We appreciate your consideration and are glad that your concerns have been addressed. We value your thoughtful evaluation.

---

### Author Response · Authors · 2024-11-14
**General response**

We would like to express our sincere gratitude for the reviewers' thoughtful feedback.

# Error bars and stability of performance improvements with priors
We have now included error bars in our experiment, and the model's performance remains stable with small variability, indicating that the observed improvements are not due to experimental randomness. The enhancement in performance is primarily driven by our proposed penalty loss, as the results show consistent gains even with error bars, confirming that the performance boost is attributable to the new loss function rather than randomness.

|  Method   | Advection  |  Burger | Shallow water | Unconditional advection
|  ---- | ----  | ----  | ----  |  ----
| w/o prior  | $1.7263 \pm 0.0491$  | $0.6862 \pm 0.0060$  |  $8.0153 \pm 0.0960$  | $0.2398 \pm 0.0024$
| w/ prior    | $1.6536 \pm 0.0677$  | $0.6610 \pm 0.0012$  |  $7.7618 \pm 0.0645$  | $0.2305 \pm 0.0001$


| Method | Three-body |  |  | Five-spring |  |  |
| :---: | :---: | :---: | :---: | :---: | :---: | :---: |
|  | Traj error | Vel error | Energy error | Dynamic error | Momentum error | Energy error |
| w/o prior | $2.4132 \pm 0.1208$ | $2.5745 \pm 0.0790$ | $4.3292 \pm 0.7235$ | $5.1754 \pm 0.0286$ | $5.3699 \pm 0.0462$ | $1.0618 \pm 0.0243$ |
| w/ prior | $1.9880 \pm 0.3418$ | $0.8328 \pm 0.1042$ | $0.5465 \pm 0.0705$ | $5.0731 \pm 0.0406$ | $0.3898 \pm 0.0118$ | $0.7418 \pm 0.0129$ |

# More ablation studies on distributional priors
We conduct additional ablation studies to investigate the importance of the distributional priors. We test two cases: 1) with data augmentation like random permutation and 2) without data augmentation. The results can be seen in the following tables.

## Three-body dataset:
| Method                          | traj error (w/o aug) | traj error (w/ aug) | vel error (w/o aug) | vel error (w/ aug) | energy error (w/o aug) | energy error (w/ aug) |
|---------------------------------|----------------------------|----------------------------|--------------------------|--------------------------|------------------------|------------------------|
| Momentum conservation           | 2.1953                     | 2.1409                     | 2.2974                   | 2.2529                   | 3.4364                 | 4.1116                 |
| Reducible energy conservation   | 1.8170                     | 1.6072                     | 1.9239                   | 0.7307                   | 3.2382                 | 0.5062                 |


## Five-spring dataset:
| Method                       | dynamic error (w/o aug) | dynamic error (w/ aug) | momentum error (w/o aug) | momentum error (w/ aug) | energy error (w/o aug) | energy error (w/ aug) |
|------------------------------|-------------------------|-------------------------|--------------------------|--------------------------|------------------------|------------------------|
| momentum conservation         | 6.3972                  | 5.0919                  | 0.3564                   | 0.3687                   | 1.5536                 | 0.7448                 |
| reducible energy conservation | 6.6211                  | 5.1615                  | 7.0716                   | 5.3032                   | 2.4910                 | 1.0548                 |

We also tested the equivalence of the trained models, one trained with data augmentation and the other without. Our results show that, even without the introduction of data augmentation for equivalence, the model still learns to exhibit the desired equivalence, which aligns with our mathematical analysis. Furthermore, when equivalence data augmentation is applied, the model achieves a significant reduction in equivalence error, decreasing from 1.6e-3 to 3.5e-4. This further supports the correctness of our analysis, demonstrating that, when using two models with the same capacity, a $\left(\mathcal{G}, \nabla^{-1}\right)$-equivariant model should be employed for noise matching.

---

### Meta-Review · Area_Chair_rND5 · 2024-12-13

**Metareview:**

The submission proposes methods for incorporating physical priors into diffusion-based generative models when used for solving physical problems (e.g., generating physical fields), which is a topic not yet heavily explored. The presented formulations seem sufficiently general and sound. The reviewers also raised a few insufficiencies, e.g., the usage of equivariant architectures and data augmentation (which may mix with the effect of the proposed loss), insignificant improvements in some cases, lack of necessary citations and discussions with relevant works, and not sufficiently diverse physical laws and more realistic demonstrations. During rebuttal, the authors have provided abundant further results and explanations, which have fixed some of the insufficiencies, and made reasonable arguments on some other points. Overall, the reviewers tend to believe the submission makes a helpful exploration on methods to build physical priors into diffusion models.

Additional personal comments: Do the formulations in Sec. 3.1 only apply to measure-preserving group operations (it is indeed the case for the two examples in the paper, though)? And the authors may be interested in a discussion on fulfilling the physical constraints by building a diffusion model directly on ECM.

**Additional Comments On Reviewer Discussion:**

Reviewer JiqC gives the only negative rating for this submission. The reviewer initially rated a clear reject, and increased the score to borderline reject after the rebuttal. His/Her initial concerns were addressed by further improvements on the paper (discussions on prior work), further results (error bars), and reasonable explanations on some of the concerns. Though not all points are fully addressed (effect of using equivariant architecture and data augmentation), the reviewer indicated that it has resolved most of the concerns. I agree with some of the concerns, but some of them do not affect the main contributions. Considering points and opinions from other reviewers, I tend to outweigh the contributions over the remaining insufficiencies.

---

### Decision · Program_Chairs · 2025-01-22

Accept (Poster)